# A PROXY MATRIX-BASED FRAMEWORK FOR CONTEXTUAL STOCHASTIC OPTIMIZATION UNDER CONFOUNDING EFFECT

## ABSTRACT

Data-driven decision-making in real-world scenarios often faces the challenge of endogeneity between decisions and outcomes, introducing confounding effects. While existing literature typically assumes unconfoundedness, this is often unrealistic. In practice, decision-making relies on high-dimensional, heterogeneous-type proxy features of confounders, leading to suboptimal decisions due to limited predictive power for uncertainty. We propose a novel semi-parametric decision framework to mitigate confounding effects. Our approach combines exponential family matrix completion to infer the confounders matrix from proxy features, with non-parametric prescriptive methods for decision-making based on the estimated confounders. We derive a non-convergent regret bound for data-driven decisions under confounding effects and demonstrate how our framework improves this bound. Experiments on both synthetic and real datasets validate our method's efficacy in reducing confounding effects across various proxy dimensions. We also show that our approach consistently outperforms benchmarks in practical applications.

## 1 INTRODUCTION

With the increasing availability of data and huge computational power combined with advancements in machine learning (ML) and optimization techniques, decision-making frameworks based on historical data have found application in a growing number of real-world scenarios (Brynjolfsson & McElheran, 2016; Sadana et al., 2024). However, in some application contexts, historical observational data may not suffice to support decision-making. A common example arises when endogeneity is present in the data, or when decisions influence uncertain outcomes. In such cases, decision-makers (DMs) can only observe past decisions and the outcomes associated with those decisions. The absence of counterfactual observations can render decision problems based on historical observational data unidentifiable (Bertsimas & Kallus, 2022), thereby making the decision-making process challenging. Such decision problems are prevalent in practice, such as pricing (Ferreira et al., 2016) and promotion (Cohen et al., 2017) strategies faced by firms like Amazon and Uber, treatment policy decisions in healthcare (Bertsimas et al., 2016), internet advertising placement optimization (Avadhanula et al., 2021), and optimization of loan interest rates (Besbes et al., 2010).

For improved decision-making, DMs typically leverage covariates, also known as contextual information or features (Mišić & Perakis, 2020). Some of these features simultaneously affect both the decision and the uncertain quantity, and are referred to as *confounders*. A commonly used approach to address the absence of counterfactual observations is to assume that all confounders can be observed, known as the *unconfoundedness* assumption. This assumption is prevalent in the literature on data-driven decision-making with endogeneity (Bertsimas & Kallus, 2020; 2022; Biggs, 2022; Biggs et al., 2021) and treatment effect research (Wager & Athey, 2018; Knaus, 2022; Armstrong & Kolesár, 2021). If this assumption is not satisfied, *confounding effects* will be present in historical data. A straightforward explanation is that, in decision-making, we need to account for how decisions influence uncertainty. However, due to the presence of confounders, inferring this influence from historical data may lead to inaccuracies, thereby resulting in misguided decisions.

The unconfoundedness assumption is untestable in reality and difficult to satisfy (Wang & Blei, 2019). First, DMs do not know which features are true confounders; they rely on experience or judgment to observe as many features as possible to avoid omissions. Second, even if DMs know certain features are confounders, they may not be able to find appropriate observations to represent them, and the observation process may introduce errors or noise, such as recording mistakes. Consequently, what DMs observe are high-dimensional *proxy features* for confounders, which may include

the true confounders or features related to them. Furthermore, high dimensionality can lead to heterogeneous data types and missing values. When using historical data containing proxy features for decision-making, we face several challenges. First, the discrepancy between proxy features and true confounders leads to confounding effects, which can impair decision-making effectiveness. Second, the high dimensionality of the data, along with heterogeneous data types and missing values, further complicates the decision-making process and affects decision performance.

In the field of causal effect research, various methods have been proposed to handle confounding effects. For instance, instrumental variables (IV) are widely used to estimate causal effects (Carrasco et al., 2007; Baiocchi et al., 2014; Chen & Qiu, 2016; Guo & Small, 2016; Mogstad & Torgovitsky, 2018). Another approach focuses on estimating causal effects using proxy features. The idea is first to understand the distributional relationship between confounders and proxies, adjust the confounders, and then identify causal effects (Wooldridge, 2009; Pearl, 2012; Cai & Kuroki, 2012; Kuroki & Pearl, 2014; Edwards et al., 2015; Miao et al., 2018; Tchetgen et al., 2020). Studies most closely related to our approach involve inferring confounders using observed proxies, typically based on latent-variable models (Kingma, 2013; Louizos et al., 2017; Kallus et al., 2018). In more complex data scenarios, other methods for inferring hidden confounders have also been developed (Guo et al., 2020; Chu et al., 2021; Ma et al., 2021). This paper focuses on addressing the confounding effect issue in stochastic optimization, providing theoretical guarantees and insights that distinguish it from casual effect literature.

The literature on decision-dependent uncertainty in the presence of contextual information is rather sparse (Bertsimas & Kallus, 2020; Bertsimas & Koduri, 2022; Bertsimas & Kallus, 2022), and studies in this area typically rely on the unconfoundedness assumption. Bertsimas & Kallus (2020) consider the impact of decisions on uncertainty for general optimization problems, while Biggs et al. (2021) and Biggs (2022) study personalized pricing problems. Unlike existing literature, which typically proceeds under the assumption of no confounding effect, we focuses on the impact when the unconfoundedness assumption is violated and explores strategies to mitigate the confounding effect. This paper investigates the impact of confounding effects in a more generalized decision-making model and utilizes proxy feature matrices and matrix-related methods to improve decision performance. Specifically, to address the challenges mentioned earlier, we propose a two-step, semi-parametric decision-making framework that integrates matrix completion and non-parametric data-driven optimization. Matrix completion methods have been extensively studied (Bennett et al., 2007; Cao et al., 2014; Schuler et al., 2016) and can be efficiently combined with causal inference techniques such as regression and matching (Imbens & Rubin, 2015; Kallus et al., 2018). We employ exponential family matrix completion (Gunasekar et al., 2014) and factorization to infer the confounders matrix. The estimated confounders matrix is then utilized in the downstream weighted sample average approximation (wSAA) framework. Our contributions are as follows:

- We extend the research of contextual stochastic optimization problems with endogeneity to scenarios where the unconfoundedness assumption does not hold. We derive a novel stochastic regret bound that quantifies the impact of confounding effects. This bound elucidates how the magnitude of confounding effects correlates with the information content of proxy features regarding the distribution of the uncertainty.

- We propose a novel semi-parametric decision-making framework, as shown in Figure 1, which is motivated by the form of the quantified confounding effect. The proposed method mitigates the inherent difficulty of satisfying unconfoundedness and reduces the confounding effect. We derive the stochastic regret bound of our proposed method and theoretically demonstrate its efficiency in reducing the confounding effect. Experiments using both synthetic and real data confirm the effectiveness of our proposed method in mitigating the confounding effect and show its superiority compared to other methods.

## 2 CONTEXTUAL OPTIMIZATION MODEL AND CONFOUNDING EFFECT

### 2.1 FULL- AND PARTIAL-INFORMATION MODELS

We consider a stochastic optimization problem where the DM seeks a decision $\boldsymbol{q} \in \mathbb{Q} \subset \mathbb{R}^p$ to maximize a utility function $\pi(\boldsymbol{q}; \mathbf{y})$ that is affected by some decision-dependent uncertainty $\mathbf{y} \in \mathbb{Y} \subset \mathbb{R}^d$. We assume that there exist confounders, denoted by $\mathbf{x} \in \mathbb{X} \subset \mathbb{R}^m$, influencing both the decision and uncertainty. The confounders $\mathbf{x}$ are interpreted as contextual information in the

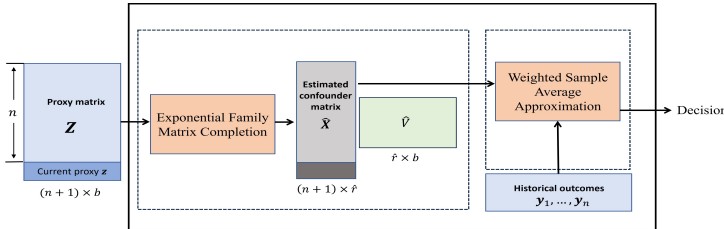

Figure 1: The structure of the proposed semi-parametric decision-making framework

decision-making scenario. For example, the uncertainty may take the form $\mathbf{y} = f(\boldsymbol{x}, \boldsymbol{q}) + \varepsilon$, where $f : \mathbb{R}^m \times \mathbb{R}^p \to \mathbb{R}^d$ is a deterministic function and $\varepsilon \in \mathbb{R}^d$ is a $d$-dimensional random vector. Given confounders $\mathbf{x} = \boldsymbol{x}$, the decision-making problem can be formulated as a contextual stochastic optimization problem:

$$[\text{FI-model}] \quad \max_{\boldsymbol{q} \in \mathbb{Q}} \mathbb{E}\left[\pi\left(\boldsymbol{q}; \mathbf{y}(\boldsymbol{q})\right) \mid \mathbf{x} = \boldsymbol{x}\right], \tag{1}$$

where the expectation is taken with respect to the conditional random variable $\mathbf{y}(\boldsymbol{q}) \mid \boldsymbol{x}$.

Problem (1) requires the knowledge of all confounders and the distribution of $\mathbf{y}(\boldsymbol{q})$ for each $\boldsymbol{q}$ conditional on $\boldsymbol{x}$. We refer to it as the Full-Information model (FI-model) and use $\boldsymbol{q}_\star$ to denote its optimal solution. Usually, instead of the confounders, only proxy features $\mathbf{z} \in \mathbb{R}^b$ can be observed, which are potentially high-dimensional with heterogeneous data types. Worse still, samples of them often contain missing values. Given proxy features $\mathbf{z} = \boldsymbol{z}$, it is natural to approximate the FI-model by the following problem:

$$[\text{PI-model}] \quad \max_{\boldsymbol{q} \in \mathbb{Q}} \mathbb{E}\left[\pi\left(\boldsymbol{q}; \mathbf{y}(\boldsymbol{q})\right) \mid \mathbf{z} = \boldsymbol{z}\right], \tag{2}$$

Problem (2) requires only the distribution $\mathbf{y}(\boldsymbol{q})$ for each $\boldsymbol{q}$ conditional on $\boldsymbol{z}$. We refer to it as the Partial-Information model (PI-model) and use $\boldsymbol{q}^*$ (different from $\boldsymbol{q}_\star$) to denote the optimal solution. Typically, confounders $\boldsymbol{x}$ contain more information about the uncertain quantity than their proxies $\boldsymbol{z}$, and the optimal expected utility of the FI-model would be higher than that of the PI-model.

## 2.2 Non-parametric optimization with unconfoundedness

Although the PI-model does not require knowing the confounders in its entirety, it relies on the distributions of $\mathbf{y}(\boldsymbol{q}) \mid \boldsymbol{z}$. Typically, these distributions are unavailable but can be probed via historical data of the form $\mathbb{D} = \{(\boldsymbol{z}_i, \boldsymbol{q}_i, \boldsymbol{y}_i)\}_{i=1,\ldots,n}$, where $\boldsymbol{z}_i$, $\boldsymbol{q}_i$, and $\boldsymbol{y}_i$ are the $i$th historical proxy features, decision, and realized outcomes, respectively. For each data point $i$, we use $\boldsymbol{x}_i$ to denote the underlined true confounders. The problem for the DM is therefore to solve the PI-model based on the sample $\mathbb{D}$.

***Unidentifiability and confounding effect.*** Ideally, if the available data is of the form $(\boldsymbol{z}_i, \boldsymbol{y}_i(\boldsymbol{q}))$ (i.e., for each $\boldsymbol{z}_i$, the function $\boldsymbol{y}_i(\boldsymbol{q})$ is observed), then the PI-model can be identified and solved by the data readily. However, the historical data $\{(\boldsymbol{z}_i, \boldsymbol{q}_i, \boldsymbol{y}_i)\}_{i=1,\ldots,n}$ follow the joint distribution of $(\mathbf{z}, \mathbf{q}, \mathbf{y})$, where the realized uncertainty $\boldsymbol{y}_i$ is the function value $\boldsymbol{y}_i(\boldsymbol{q}_i)$ under the input $\boldsymbol{q}_i$. The function $\boldsymbol{y}(\boldsymbol{q})$ and the optimal decision are unidentifiable from the observed data, as the historical decisions and outcomes are confounded. The unidentifiability and confounding effect would lead to sub-optimal decisions. The typical approach to address these two issues simultaneously is to have the following unconfoundedness assumption (also known as exogeneity, ignorability, and conditional independence).

**Assumption 1 (Unconfoundedness)** *For every* $\boldsymbol{q} \in \mathbb{Q}$, $\mathbf{y}(\boldsymbol{q}) \perp \mathbf{q} \mid \mathbf{z}$.

Assumption 1 means that the proxy features contain all confounders, which further implies that conditioned on $\boldsymbol{z}_i$, the historical decision $\boldsymbol{q}_i$ is statistically independent of the uncertainty $\boldsymbol{y}_i$, or they are as-if random. This assumption also implies that the proxy features include all factors that simultaneously affect both decision and uncertainty.

With Assumption 1, the PI-model can be reformulated as

$$\max_{\boldsymbol{q}\in\mathbb{Q}} \mathbb{E}\left[\pi\left(\boldsymbol{q};\mathbf{y}(\boldsymbol{q})\right) \mid \mathbf{z}=\boldsymbol{z}\right] = \max_{\boldsymbol{q}\in\mathbb{Q}} \mathbb{E}\left[\pi\left(\boldsymbol{q};\mathbf{y}\right) \mid \mathbf{z}=\boldsymbol{z}, \mathbf{q}=\boldsymbol{q}\right], \tag{3}$$

where the objective function depends on the joint distribution of $(\mathbf{z}, \mathbf{q}, \mathbf{y})$. Therefore, the historical data is sufficient to identify this problem and the confounding effect no longer exists.

Now we turn to solve the problem (3) based on the dataset $\mathbb{D}$. Since the functional form of uncertainty is unknown, one popular and effective non-parametric data-driven method is the wSAA approach proposed by Bertsimas & Kallus (2020), which is a variant of the SAA approach with each data point weighted according to the similarity between the current decision-proxy pair and the historical decision-proxy pair. Specifically, the weight assigned to the $i$th data, $w_i(\boldsymbol{z}, \boldsymbol{q})$, is a function that measures the closeness between vectors $(\boldsymbol{z}, \boldsymbol{q})$ and $(\boldsymbol{z}_i, \boldsymbol{q}_i)$. One of the typical weight functions is defined by the $k$-nearest neighbor ($k$NN) and presented below. For more weight functions, please refer to Bertsimas & Kallus (2020).

**Definition 1** *The kNN weight functions are given by*

$$w_i(\boldsymbol{z}, \boldsymbol{q}) = \frac{1}{k}\mathbb{1}[(\boldsymbol{z}_i, \boldsymbol{q}_i) \text{ is a kNN of } (\boldsymbol{z}, \boldsymbol{q})], \tag{4}$$

*where $k = \lceil cn^\gamma \rceil$, $\gamma \in (0,1)$, $c > 0$, $(\boldsymbol{z}_i, \boldsymbol{q}_i)$ is a k-nearest neighbor of $(\boldsymbol{z}, \boldsymbol{q})$ if $|\{i_* \in \{1, ..., N\}\setminus i : dist((\boldsymbol{z}_{i_*}, \boldsymbol{q}_{i_*}), (\boldsymbol{z}, \boldsymbol{q})) < dist((\boldsymbol{z}_i, \boldsymbol{q}_i), (\boldsymbol{z}, \boldsymbol{q}))\}| < k$.*

Adopting the wSAA approach with this weight function, a data-driven PI-model (DDPI-Model) is:

$$[\text{DDPI-model}] \max_{\boldsymbol{q}\in\mathbb{Q}} \check{\mathbb{E}}\left[\pi\left(\boldsymbol{q};\mathbf{y}\right) \mid \mathbf{z}=\boldsymbol{z}, \mathbf{q}=\boldsymbol{q}\right] = \sum_{i=1}^{n} w_i(\boldsymbol{z}, \boldsymbol{q})\pi(\boldsymbol{q}; \boldsymbol{y}_i). \tag{5}$$

Let $\check{\boldsymbol{q}}$ denote the optimal data-driven solution for DDPI-model. This paper does not focuses on the computational aspect of problem (5). We however emphasize that $\check{\boldsymbol{q}}$ can be computed in polynomial time (Bertsimas & Kallus, 2020), with efficient algorithms established (Liu & Zhang, 2023).

***Performance of the DDPI-model.*** We demonstrate under Assumption 1 that the DDPI-model effectively approximates the PI-model. Given proxy features $\boldsymbol{z}$, the regret of the DDPI-model is defined as the gap between the optimal expected utility and the expected utility of the DDPI solution $\check{\boldsymbol{q}}$:

$$\mathcal{R}_1(\boldsymbol{z}) = \mathbb{E}\left[\pi\left(\boldsymbol{q}^*;\mathbf{y}\right) \mid \boldsymbol{z}, \boldsymbol{q}^*\right] - \mathbb{E}\left[\pi\left(\check{\boldsymbol{q}};\mathbf{y}\right) \mid \boldsymbol{z}, \check{\boldsymbol{q}}\right]. \tag{6}$$

To derive the stochastic bound of this regret, we decompose $\boldsymbol{q}$ into $\boldsymbol{q}_1$ (affecting the uncertainty distribution) and $\boldsymbol{q}_2$ (influencing the utility function, independent of uncertainty). This decomposition facilitates the proofs and allows for a weaker assumption on $\boldsymbol{q}$.

**Assumption 2 (Utility Function)** *(i) $\mathbb{Q}$ is a compact set. (ii) For any $\boldsymbol{y} \in \mathbb{Y}$, $\boldsymbol{q} \mapsto \pi(\boldsymbol{q};\mathbf{y})$ is $L(\boldsymbol{y})$-Lipschitz continuous on $\mathbb{Q}$. Moreover, for any $(\boldsymbol{z}, \boldsymbol{q}_1) \in \mathbb{A}$, $\mathbb{E}\left[L(\boldsymbol{y}) \mid \boldsymbol{z}, \boldsymbol{q}_1\right] \leq L_1 < \infty$. (iii) For any $\boldsymbol{q}_2 \in \mathbb{Q}$, $(\boldsymbol{z}, \boldsymbol{q}_1) \mapsto \mathbb{E}\left[\pi(\boldsymbol{q}_2;\mathbf{y}) \mid \boldsymbol{z}, \boldsymbol{q}_1\right]$ is $L_2$-Lipschitz continuous on $\mathbb{Z} \times \mathbb{Q}$. (iv) $(\boldsymbol{z}, \boldsymbol{q}_1)$ has a support $\mathbb{A} \subset [0,1]^{b+p}$, and there exists $g > 0$ such that $\mathbb{P}\left\{(\mathbf{z}, \boldsymbol{q}_1) \in B_\epsilon(\boldsymbol{z}, \boldsymbol{q}_1)\right\} > g\epsilon^{b+p}$ for any $(\boldsymbol{z}, \boldsymbol{q}_1) \in \mathbb{A}$ and $\epsilon > 0$. (v) For any $\boldsymbol{q}_2 \in \mathbb{Q}$ and $(\boldsymbol{z}, \boldsymbol{q}_1) \in \mathbb{A}$, the random variable $\pi(\boldsymbol{q}_2;\mathbf{y}) - \mathbb{E}\left[\pi(\boldsymbol{q}_2;\mathbf{y}) \mid \boldsymbol{z}, \boldsymbol{q}_1\right]$ is sub-Gaussian with variance proxy $\sigma_1^2$ independent of $(\boldsymbol{z}, \boldsymbol{q}_1)$ and $\boldsymbol{q}_2$. (vi) Conditions (iii)–(v) still hold when proxy features $\boldsymbol{z}$ are replaced by confounders $\boldsymbol{x}$.*

Conditions (i)-(iii) are standard assumptions in contextual optimization literature (Sadana et al., 2024; Bertsimas & McCord, 2019; Lin et al., 2022). Condition (iv) is satisfied for finite support of $(\mathbf{z}, \mathbf{q})$. Condition (v) is a technical assumption for bound derivation. Conditions (iv) and (v) align with settings in Bertsimas & McCord (2019) and Rahimian & Pagnoncelli (2023). The last condition parallels conditions (iii), (iv), and (v). Then we have the following result.

**Theorem 1 (Stochastic Regret bound of the DDPI-model)** *Suppose Assumptions 1 and 2 hold, and the samples $\mathbb{D}$ are i.i.d.. Then, under the kNN weight functions (4), for any confidence level $\alpha \in (0,1)$ and $n \geq 2p + 2b$, with probability at least $1 - \alpha$, we have $\mathcal{R}_1(z) \leq \xi_{\alpha,p,b}$, where*

$$\xi_{\alpha,p,b} = \max\left\{ n^{-\frac{1}{2p+2b}} C_1 \left(\frac{2p+b}{2p+2b}\log n + C_2\right)^{\frac{1}{2p+2b}}, n^{-\frac{\gamma}{2}} C_3 \left(\sqrt{\frac{4p+4b+\gamma p}{2}\log n} + C_4\right)\right\},$$

$C_1$, $C_2$, $C_3$ and $C_4$ *are constants defined in Appendix A.1 and* $C_1$, $C_2$ *and* $C_4$ *depend on $p$ and $b$.*

Theorem 1 establishes the regret bound of the DDPI-model under the unconfoundedness assumption as $\tilde{\mathcal{O}}\left(n^{-\frac{1}{2p+2b}}\right)$ with high probability, with $p$ being the decision dimension and $b$ the number of proxy features. This result implies that as $n$ increases, the regret in (6) approaches zero with high probability, and the convergence rate decreases as the number of proxy features increases. Moreover, the DDPI-model achieves asymptotically optimal performance in estimating the PI-model.

## 2.3 UTILITY GAP CAUSED BY CONFOUNDING EFFECT

Despite the DDPI-model's effectiveness, real-world scenarios introduce additional complexities. Assumption 1, which posits absence of confounding effects, is usually violated. The discrepancy between observed proxy features $\boldsymbol{z}$ and ideal confounders $\boldsymbol{x}$ inevitably introduces confounding effects. In the idealized scenario where confounders $\boldsymbol{x}$ are directly observed, we define the Data-Driven FI-model (DDFI-model) as:

$$[\text{DDFI-model}] \ \max_{\boldsymbol{q}\in\mathbb{Q}} \tilde{\mathbb{E}}\left[\pi\left(\boldsymbol{q};\mathbf{y}\right)\mid\boldsymbol{x},\boldsymbol{q}\right] = \sum_{i=1}^{n} w_i(\boldsymbol{x},\boldsymbol{q})\pi(\boldsymbol{q};\boldsymbol{y}_i). \tag{7}$$

Let $\tilde{\boldsymbol{q}}$ denote the optimal solution of the DDFI-model. Given that $\boldsymbol{x}$ contains more information about $\mathbf{y}$ compared with its proxy features $\boldsymbol{z}$, the DDFI-model is expected to outperform the DDPI-model. The confounding effects will cause the performance gap between these models, as shown in the numerical visualization in Section 5.1. Theoretically, we quantify the confounding effect as the utility gap between the FI-model's optimal expected utility and the expected utility from the DDPI-model's data-driven solution $\check{\boldsymbol{q}}$. This 'true' regret of the DDPI-model is expressed as:

$$\mathcal{R}_2(\boldsymbol{x},\boldsymbol{z}) = \mathbb{E}\left[\pi\left(\boldsymbol{q}_\star;\mathbf{y}\right)\mid\boldsymbol{x},\boldsymbol{q}_\star\right] - \mathbb{E}\left[\pi\left(\check{\boldsymbol{q}};\mathbf{y}\right)\mid\boldsymbol{x},\check{\boldsymbol{q}}\right]. \tag{8}$$

**Theorem 2 (Regret Bound Caused by Confounding Effect)** *Suppose the conditions in Theorem 1 hold. Then, the regret defined in (8) satisfies*

$$\mathcal{R}_2(\boldsymbol{x},\boldsymbol{z}) \le \mathcal{R}_1(\boldsymbol{z}) + 2\sup_{\boldsymbol{q}} \left|\mathbb{E}\left[\pi\left(\boldsymbol{q};\mathbf{y}\right)\mid\boldsymbol{x},\boldsymbol{q}\right] - \mathbb{E}\left[\pi\left(\boldsymbol{q};\mathbf{y}\right)\mid\boldsymbol{z},\boldsymbol{q}\right]\right|. \tag{9}$$

*Suppose additionally that $b = m$. Then, for any confidence level $\alpha \in (0,1)$ and $n \ge 2p + 2b$, with probability at least $1 - \alpha$, we have*

$$\mathcal{R}_2(\boldsymbol{x},\boldsymbol{z}) \le \xi_{\alpha,p,b} + 2L_2\|\boldsymbol{x} - \boldsymbol{z}\|. \tag{10}$$

Theorem 2 provides a theoretical result regarding the confounding effect. The first part of Theorem 2 shows that the upper bound of the confounding effect comprises two terms: (i) the DDPI-model's regret, which converges to 0 at $\tilde{\mathcal{O}}\left(n^{-\frac{1}{2p+2b}}\right)$ (from Theorem 1); (ii) utility gap between FI- and PI-models uniformly over decision $\boldsymbol{q}$, which depends on information about $\mathbf{y}$ contained in $\boldsymbol{z}$, and independent of sample size. The second term approaches zero only when $\boldsymbol{z}$ fully captures information in confounders $\boldsymbol{x}$. The second part of Theorem 2 shows that when the numbers of confounders and proxy features are the same (i.e., $b = m$), the second term in (9) is expressible as the Euclidean norm $\|\boldsymbol{x} - \boldsymbol{z}\|$. This term is independent of the sample size and will only equal zero when the proxy features are exactly the same as the confounders. Theorem 2 highlights that when the unconfoundedness assumption is violated, the confounding effect leads to sub-optimal performance of the DDPI-model in practical applications.

## 3 PROXY MATRIX-BASED DECISION-MAKING FRAMEWORK

The confounding effect arises from the discrepancy in predictiveness between proxy features and confounders. Intuitively, the more information about $\boldsymbol{x}$ the proxy features $\boldsymbol{z}$ contain, the tighter the bound in (9) is. This insight motivates our two-step, semi-parametric decision-making framework. We first estimate the confounders matrix from the proxy matrix using matrix completion and factorization techniques, then make decisions using the wSAA framework based on the inferred confounders. Such matrix-related methods enjoy three advantages. (i) Versatility: Effective in handling high-dimensional, heterogeneous data with missing values. (ii) Compatibility: demonstrated synergy with causal inference methods (Kallus et al., 2018). (iii) Theoretical guarantees: provide rigorous mathematical guarantees.

## 3.1 STRUCTURE OF PROXY MATRIX

Let $\overline{\boldsymbol{Z}} = (\boldsymbol{z}_1, ..., \boldsymbol{z}_n)^\top \in \mathbb{R}^{n \times b}$ represent historical proxy features, where $b$ is the feature dimension. Denote the current observed feature vector as $\boldsymbol{z}$. We construct $\boldsymbol{Z} = (\boldsymbol{z}_1, ..., \boldsymbol{z}_n, \boldsymbol{z})^\top \in \mathbb{R}^{(n+1) \times b}$ by appending the row vector $\boldsymbol{z}^\top$ to the bottom of $\overline{\boldsymbol{Z}}$. Similarly, we obtain $\boldsymbol{X} = (\boldsymbol{x}_1, ..., \boldsymbol{x}_n, \boldsymbol{x})^\top \in \mathbb{R}^{(n+1) \times m}$ for confounders, with the last row corresponding to current true confounders.

***Proxy matrix generation.*** There exists a loading matrix $\boldsymbol{V} \in \mathbb{R}^{m \times b}$ such that the proxy matrix is generated by

$$\boldsymbol{Z} \sim \mathbb{P}(\boldsymbol{Z}|\boldsymbol{A}), \quad \boldsymbol{A} = \boldsymbol{X}\boldsymbol{V} = \begin{pmatrix} X_{1,1} & X_{1,2} & \cdots & X_{1,m} \\ X_{2,1} & X_{2,2} & \cdots & X_{2,m} \\ \vdots & \vdots & \ddots & \vdots \\ X_{n+1,1} & X_{n+1,2} & \cdots & X_{n+1,m} \end{pmatrix} \begin{pmatrix} V_{1,1} & V_{1,2} & \cdots & V_{1,b} \\ V_{2,1} & V_{2,2} & \cdots & V_{2,b} \\ \vdots & \vdots & \ddots & \vdots \\ V_{m,1} & V_{m,2} & \cdots & V_{m,b} \end{pmatrix}.$$

For any proxy feature $Z_{i,j}$ (or $z_{ij}$) with $i \in [n+1]$ and $j \in [b]$, we have $Z_{i,j} \sim \mathbb{P}\left(\cdot | \sum_{s=1}^{m} X_{i,s} V_{s,j}\right)$. This formulation posits $Z_{i,j}$ as a noisy realization of a linear combination of the $i$th confounders (via distribution $\mathbb{P}(\cdot)$). $V_{s,j}$ quantifies the extent to which the $s$th confounder is proxied by the $j$th feature, with larger values indicating stronger informational content. While we present a linear relationship, this formulation is not restrictive. Nonlinear underlying relationships can be approximated by a linear form in an expanded feature space through Taylor approximation.

## 3.2 THE TWO-STEP DECISION MAKING

***Step 1. M-Estimator for confounders.*** Based on the observed proxy features matrix $\boldsymbol{Z}$, we employ M-estimation (Gunasekar et al., 2014) to estimate matrix $\boldsymbol{A}$ based on the proxy matrix:

$$\min_{\|\boldsymbol{A}\|_\infty \leq \frac{\gamma}{\sqrt{(n+1)b}}} \sum_{i,j \in \Omega} -\frac{(n+1)b}{|\Omega|} \log \mathbb{P}\left(Z_{i,j}|A_{i,j}\right) + \lambda \|\boldsymbol{A}\|_\star, \tag{11}$$

where $\|\cdot\|_\star$ is the nuclear norm, $\lambda > 0$ is a tuning parameter, and $\Omega$ is the subset of indices $[n+1] \times [b]$ for which the proxy features are observed (refer to Assumption 5 for details). We denote the M-estimator as $\hat{\boldsymbol{A}}$, i.e., the optimal solution of problem (11). Note that a larger $\lambda$ implies a lower rank of $\hat{\boldsymbol{A}}$. The constraint is only for technical use and can be ignored in practice. For a sufficiently large $\hat{r}$, problem (11) is equivalent to the following problem (Kallus & Udell, 2020), which is computationally more tractable:

$$\min_{\boldsymbol{X} \in \mathbb{R}^{(n+1) \times \hat{r}}, \boldsymbol{V} \in \mathbb{R}^{\hat{r} \times b}} \sum_{i,j \in \Omega} -\frac{(n+1)b}{|\Omega|} \log \mathbb{P}\left(Z_{i,j}| \sum_{s=1}^{\hat{r}} X_{i,s} V_{s,j}\right) + \frac{\lambda}{2} \|\boldsymbol{X}\|_F^2 + \frac{\lambda}{2} \|\boldsymbol{V}\|_F^2, \tag{12}$$

where $\|\cdot\|_F$ denotes the Frobenius norm, $\hat{r}$ is the estimated rank of $\boldsymbol{X}$ determined by the cross-validation. The optimal solution $\hat{\boldsymbol{X}} \in \mathbb{R}^{(n+1) \times \hat{r}}$ is equivalent to the left singular matrix of $\hat{\boldsymbol{A}}$. Thus $\hat{\boldsymbol{X}}$ has orthogonal columns, and any non-singular linear transformation of $\hat{\boldsymbol{X}}$ can serve as a valid estimator of the confounder matrix. For simplicity, we use $\hat{\boldsymbol{X}}$ in subsequent decision making.

***Step 2. Decision-making based on estimated confounders.*** The estimated confounders matrix can be denoted by $\hat{\boldsymbol{X}} = (\hat{\boldsymbol{x}}_1, ..., \hat{\boldsymbol{x}}_n, \hat{\boldsymbol{x}})^\top \in \mathbb{R}^{(n+1) \times \hat{r}}$, where the first $n$ rows are the estimated historical confounders (i.e., $\hat{\boldsymbol{x}}_1, ..., \hat{\boldsymbol{x}}_n$) and the last row represents the estimated current confounders (i.e., $\hat{\boldsymbol{x}}$). Subsequently, we use the estimated confounders matrix and historical data to apply the wSAA framework. The optimization problem can be formulated as:

$$[\text{PMFI-model}] \max_{\boldsymbol{q} \in \mathbb{Q}} \hat{\mathbb{E}}\left[\pi\left(\boldsymbol{q}; \mathbf{y}\right) \mid \hat{\boldsymbol{x}}, \boldsymbol{q}\right] = \sum_{i=1}^{n} w_i(\hat{\boldsymbol{x}}, \boldsymbol{q}) \pi(\boldsymbol{q}; \boldsymbol{y}_i), \tag{13}$$

where the weights $w_i(\hat{\boldsymbol{x}}, \boldsymbol{q})$ for $i = 1, ...n$ is to measure the similarity between the combination of estimated confounders and the decision (i.e., $(\hat{\boldsymbol{x}}, \boldsymbol{q})$) and the combination of $i$th estimated historical confounders and historical decision (i.e., $(\hat{\boldsymbol{x}}_i, \boldsymbol{q}_i)$). Problem (13) involves approximating the DDFI-model using methods based on the proxy matrix, named as the Proxies Matrix-Based DDFI-model (PMFI-model) with optimal decision $\hat{\boldsymbol{q}}$.

## 4 THEORETICAL GUARANTEE

We measure the performance by the difference between the FI-model's optimal expected utility and the expected utility obtained from the data-driven solution of the PMPI-model. This utility gap also represents the regret of the PMPI-model, defined as follows:

$$\mathcal{R}_3(\boldsymbol{x}, \hat{\boldsymbol{x}}) = \mathbb{E}\left[\pi\left(\boldsymbol{q}_\star; \mathbf{y}\right) \mid \boldsymbol{x}, \boldsymbol{q}_\star\right] - \mathbb{E}\left[\pi\left(\hat{\boldsymbol{q}}; \mathbf{y}\right) \mid \boldsymbol{x}, \hat{\boldsymbol{q}}\right]. \tag{14}$$

Before presenting our further results, we require the following technical assumptions.

**Assumption 3 (Proxy Generation)** *There exist functions $h : \mathbb{R} \to \mathbb{R}$ and $G : \mathbb{R} \to \mathbb{R}$ such that $G$ is strictly convex and analytic, $\nabla^2 G(u) \geq e^{-\eta|u|} \ \forall u \in \mathbb{R}$ for some $\eta > 0$, and for any $(i, j) \in [n+1] \times [b]$, the proxy feature $Z_{i,j}$ is drawn from $\mathbb{P}(Z_{i,j}|A_{i,j}) = h(Z_{i,j}) \exp(Z_{i,j} A_{i,j} - G(A_{i,j}))$.*

The proxy features can be generated by any distribution in the natural exponential family, like Gaussian, Bernoulli, Binomial, Poisson, and Exponential. The conditions about the function $G$ are also satisfied for these distributions (Gunasekar et al., 2014). Therefore, different types of proxy variables found in practice can be generated through this process.

**Assumption 4 (Matrix Conditions)** *(i) $\boldsymbol{A}$ is of rank $r = o(n)$. (ii) $\alpha_{sp}(\boldsymbol{A}) \triangleq \frac{\sqrt{(n+1)b}\|\boldsymbol{A}\|_{\max}}{\|\boldsymbol{A}\|_F} = \mathcal{O}(1)$. (iii) $\sigma_{\max}(\boldsymbol{A}) = \mathcal{O}(1)$. (iv) $\|\hat{\boldsymbol{x}} - \boldsymbol{x}\|^2 = \mathcal{O}\left(\frac{\|\hat{\boldsymbol{X}} - \boldsymbol{X}\boldsymbol{B}^*\|_F^2}{n}\right)$ where $\boldsymbol{B}^* = \arg\inf_{\boldsymbol{B} \in \mathbb{O}_r} \|\hat{\boldsymbol{X}} - \boldsymbol{X}\boldsymbol{B}\|_F^2$ and $\mathbb{O}_r = \{\boldsymbol{M} \in \mathbb{R}^{r \times r} : \boldsymbol{M}^\top \boldsymbol{M} = \boldsymbol{I}_r\}$ is the set of $r \times r$ orthogonal matrices.*

In Assumption 4, the big-O (and small-O) notations are with respect to $n$ and $b$. $\sigma_{\max}(\boldsymbol{A})$ is the largest singular value of $\boldsymbol{A}$. Conditions (i)-(iii) ensure the estimation accuracy of $\boldsymbol{A}$ and the confounders matrix, while condition (iv) guarantees that an accurate estimate of the confounders matrix translates into an accurate estimate of the current confounders.

**Assumption 5 (Uniform Observations)** *Indices in $\Omega$ are i.i.d. samples of Uniform$([n+1] \times [b])$.*

This assumption is common for the low-rank structure matrix with missing values (Gunasekar et al., 2014), which means that we observe proxy features follow a uniform sampling model.

To ease analysis, we first consider a special case where confounders are known but observed through noisy one-to-one proxy features. For example, if product quality is a confounder, DMs would use customer ratings as the sole proxy feature. This case also implies that $b = m = \hat{r}$ and $\boldsymbol{V} = \boldsymbol{I}_m$.

**Theorem 3 (Regret Bound with One-to-One Realization of Proxy)** *Suppose Assumptions 2-5 hold, $Z_{i,j} - g(X_{i,j})$ is sub-Gaussian with variance proxy $\sigma_2^2$, and $|\Omega| > c_0 rm \log m$. Let $\lambda = 2c_1\sigma_2\sqrt{(n+1)m}\sqrt{\frac{rm \log m}{|\Omega|}}$. Then, under the kNN weight functions (4), for any $0 < \alpha < 1 - c_2 e^{-c_3 \log m}$ and $n \geq 2p + 2m$, with probability at least $1 - \alpha - c_2 e^{-c_3 \log m}$, we have*

$$\mathcal{R}_3(\boldsymbol{x}, \hat{\boldsymbol{x}}) \leq \xi_{\alpha,p,m} + \zeta_{m,r}, \tag{15}$$

*where $g(\boldsymbol{X}) = \nabla G(\boldsymbol{X}) \in \mathbb{R}^{(n+1) \times b}$, $\zeta_{m,r} \triangleq \frac{c_4 \alpha_{sp}(\boldsymbol{A}) \sigma_{\max}(\boldsymbol{A}) \sigma_2}{e^{-2\eta \|\boldsymbol{A}\|_{\max}}} \sqrt{\frac{r^3 m^2 \log m}{|\Omega|}}$, $c_0$, $c_1$, $c_2$, $c_3$ are positive constants and $c_4$ is a constant defined in Appendix A.2.*

Theorem 3 states that when there is a one-to-one correspondence between the confounders and proxy features, the regret is $\tilde{\mathcal{O}}\left(n^{-\frac{1}{2p+2m}} \bigvee \sqrt{\frac{r^3 m^2 \log m}{|\Omega|}}\right)$ with high probability. As $r \ll n$ and the number of proxy features $m$ is fixed, this regret will converge to 0 as the sample size $n \to \infty$ and the support satisfies $\frac{r^3 m^2 \log m}{|\Omega|} \to 0$. To ensure the regret's convergence, the number of missing values in the proxy features should not explode with $n$. As the dimension of the proxy features is fixed ($b = m$), the DMs can enhance the model's performance by increasing the sample size. Moreover, a greater number of confounders requires a larger sample to achieve improved performance.

**Theorem 4 (Regret Bound of the PMFI-model)** *Suppose Assumptions 2-5 hold. Then, under the kNN weight functions (4), we have*

$$\mathcal{R}_3(\boldsymbol{x}, \hat{\boldsymbol{x}}) \leq \mathcal{R}_1(\hat{\boldsymbol{x}}) + 2 \sup_{\boldsymbol{q} \in \mathbb{Q}} |\mathbb{E}\left[\pi\left(\boldsymbol{q}; \mathbf{y}\right) \mid \boldsymbol{x}, \boldsymbol{q}\right] - \mathbb{E}\left[\pi\left(\boldsymbol{q}; \mathbf{y}\right) \mid \hat{\boldsymbol{x}}, \boldsymbol{q}\right]|. \tag{16}$$

*Suppose additionally that $\hat{r} = m$, $\forall (i,j)$, $Z_{i,j} - g(X_{i,j})$ is sub-Gaussian with variance proxy $\sigma_2^2$, $|\Omega| > c_0 rb \log b$ and $\lambda = 2c_1 \sigma_2 \sqrt{(n+1)b} \sqrt{\frac{rb \log b}{|\Omega|}}$. Then, for any $0 < \alpha < 1 - c_2 e^{-c_3 \log b}$ and $n \geq 2p + 2m$, with probability at least $1 - \alpha - c_2 e^{-c_3 \log b}$, we have*

$$\mathcal{R}_3(\boldsymbol{x}, \hat{\boldsymbol{x}}) \leq \xi_{\alpha,p,b} + \varphi_{b,r}, \tag{17}$$

*where $\varphi_{b,r} \triangleq \frac{c_5 r \zeta_{b,r}}{\sigma_{\min}(\hat{\boldsymbol{A}})}$ and $c_5$ is a constant defined in Appendix A.2.*

The first term of (16) is due to the limitations of the non-parametric decision approach, which is analogous to the regret defined in (6), with estimated confounders $\hat{\boldsymbol{x}}$ replacing proxy features $\boldsymbol{z}$. This term is bounded by $\tilde{\mathcal{O}}\left(n^{-\frac{1}{2p+2\hat{r}}}\right)$. Given that $\hat{r} < b$, $\mathcal{R}_1(\hat{\boldsymbol{x}})$ converges faster than $\mathcal{R}_1(\boldsymbol{z})$ in (9). As $n$ and $b$ increase, $\hat{\boldsymbol{A}}$ approaches the true matrix, indicating that $\hat{\boldsymbol{x}}$ captures confounder and uncertainty information more accurately than $\boldsymbol{z}$. Consequently, the second term in (16) caused by the data generating process is expected to be smaller when conditioned on $\hat{\boldsymbol{x}}$ versus $\boldsymbol{z}$. In the second part of Theorem 4, when $\hat{r} = m$ (the estimated number of confounders equals the true value), $\varphi_{b,r}$ in (17) is $\mathcal{O}\left(\sqrt{\frac{r^5 b^2 \log b}{|\Omega|}}\right)$. This term converges to 0 as $\frac{r^5 b^2 \log b}{|\Omega|} \to 0$ when $n \to \infty$ or $b \to \infty$. We can also interpret the expression $\frac{|\Omega|}{r^5 b^2 \log b}$ as the quality of proxy features, where higher quality leads to better model performance and lower regret. Thus, the regret in (17) is upper bounded by $\tilde{\mathcal{O}}\left(n^{-\frac{1}{2p+2m}} \bigvee \sqrt{\frac{r^5 b^2 \log b}{|\Omega|}}\right)$. This bound significantly improves upon (10) in two key aspects: (i) convergence: our bound is convergent, whereas (10) reaches zero only under perfect confounder-proxy alignment; (ii) controllability: the regret can be reduced by adjusting $n$ and $b$, offering practical avenues for improvement.

## 5 NUMERICAL EXPERIMENTS

In our synthetic and real data experiments, we take the classic joint inventory and pricing problem as the contextual stochastic optimization example for evaluation. This problem simultaneously optimizes product prices and order quantities to maximize profit, with pricing decisions directly impacting uncertain demand. The problem formulation is detailed in Appendix B, while dataset specifications are provided in Appendix C.

### 5.1 VISUALIZATIONS OF THE UTILITY GAP REDUCTION

We first visualize the confounding effect by comparing the performance (i.e., profit) of the DDFI-model against DDPI-models with varying dimensions of proxy features, based on synthetic datasets. Figure 2 shows a significant gap between the average out-of-sample performance of the DDPI-model and the DDFI-model, regardless of the number of proxy features $b$ and weight functions ($k$NN and random forest). This gap represents the discrepancy in predictive capability between proxy features and confounders, effectively quantifying the utility gap induced by confounding effects. While Figure 2 suggests that DMs can partially mitigate confounding effects by adjusting the dimension of proxy features, the impact of this approach is still limited.

We also visualize the efficacy of the proposed solution method in mitigating confounding effect. We implement our proposed semi-parametric decision-making framework across various proxy feature numbers ($b = 50, 100, 500$) and assess the out-of-sample performance in Figure 3. The dimension of the estimated confounders is determined through cross-validation. The PMFI-model is shown to significantly improve out-of-sample performance, irrespective of the dimension of proxy features used. It aligns with our theoretical analysis in Theorem 4. Specifically, it supports our hypothesis that our proposed PMPI-model, leveraging M-estimation techniques, extracts more accurate confounder information and thus enhances demand prediction accuracy and decision-making quality.

### 5.2 AN REAL-WORLD RETAILING APPLICATION

We test the proposed framework on sales data from AEON, which is one of the largest retail companies in Japan. In the main text, we present the results using data from a specific store. In Appendix C, we further demonstrate the experimental outcomes with other stores' datasets to illustrate the ro-

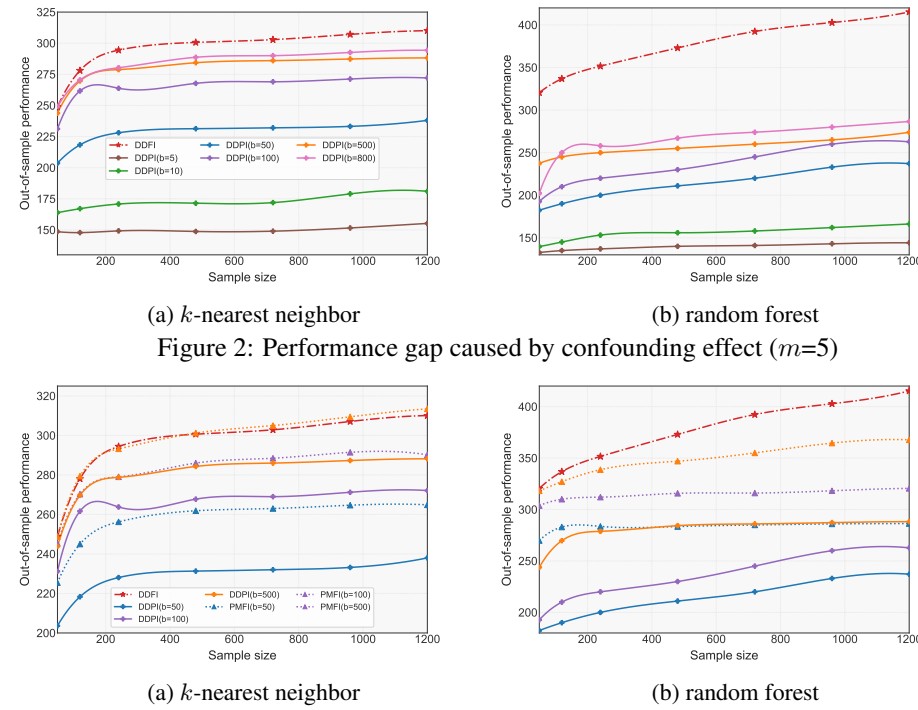

(a) $k$-nearest neighbor

(b) random forest

Figure 2: Performance gap caused by confounding effect ($m$=5)

(a) $k$-nearest neighbor

(b) random forest

Figure 3: Performance gap reduction by the PMFI-model ($m$=5)

bustness of the proposed method. We benchmark our proposed semi-parametric framework against established methodologies:

- First-predict-then-optimize (FPTO) framework (El Balghiti et al., 2019; Elmachtoub et al., 2020). FPTO framework involves two steps: (i) predicting uncertainty based on proxy features; (ii) making decisions based on predictive outcomes. We consider the following predictive models: (a) parametric methods: Ordinary Least Squares (OLS), LASSO regression (Hara & Maehara, 2017; Muthukrishnan & Rohini, 2016); (b) non-parametric methods: random forest (RF), $k$NN; and (c) deep learning: deep neural networks (DNNs), for their ability to handle high-dimensional data (Yu & Yan, 2020; Han et al., 2023).
- First-select-then-optimize (FSTO) framework. FSTO framework is designed for high-dimensional problems, including two steps: (i) feature selection; (ii) decision-making using wSAA with RF and $k$NN weight functions based on the selected features. We employ two feature selection methods: (a) LASSO regression; (b) knockoff filter: a state-of-the-art statistical technique (El Balghiti et al., 2019; Elmachtoub et al., 2020). These methods are denoted as LASSO-RF ($k$NN) and KNOCKOFF-RF ($k$NN), respectively.
- First-learning-then-optimize (FLTO) framework. FSTO framework includes two steps: (i) confounders learning; (ii) decision-making using wSAA. We employ three representation learning methods to learn confounders from proxy features: (a) Principal Components Analysis (PCA); (b) Balancing Neural Network (BNN) (Johansson et al., 2016); (c) Variational Autoencoders (VAEs) (Kingma, 2013; Louizos et al., 2017).

All hyperparameters are tuned via cross-validation. Table 1 presents the out-of-sample performance comparison across varying proxy feature dimensions. We consider evaluation metrics: out-of-sample profit and decision accuracy, measured by the mean square error (MSE) between the resulting inventory decisions and the actual demand. Our semi-parametric solution methods (PMFI-RF and PMFI-$k$NN) outperform benchmark methods in terms of out-of-sample profit and decision accuracy, underscoring the efficacy of matrix-related methods in confounder estimation and their successful integration with the wSAA framework.

Figure 4 illustrates the close alignment between resulting inventory decisions and actual demands for products in the test set. This demonstrates our method's effectiveness in extracting confounder information for decision-making. Figure 5 showcases the asymptotic properties of PMFI-RF and

Table 1: Out-of-sample performance comparison: PMFI Models vs. Benchmarks

| Method | Out-of-sample profit | | | Avg | Decision accuracy (MSE, $b$=438) |
|---|---|---|---|---|---|
| | $b$=146 | $b$=292 | $b$=438 | | |
| FPTO-OLS | 14.17 | 18.12 | 10.86 | 14.38 | 2149.78 |
| FPTO-LASSO | 10.55 | 16.15 | 16.10 | 14.27 | 456.36 |
| FPTO-RF | 20.04 | 20.49 | 21.22 | 20.87 | 405.80 |
| FPTO-$k$NN | 18.72 | 18.00 | 18.04 | 18.25 | 295.80 |
| FPTO-DNNs | 17.35 | 18.58 | 18.72 | 18.22 | 489.73 |
| LASSO-RF | 12.83 | 11.39 | 17.23 | 13.82 | 572.74 |
| LASSO-$k$NN | 12.76 | 9.94 | 11.82 | 11.51 | 358.54 |
| KNOCKOFF-RF | 17.35 | 17.45 | 17.81 | 17.54 | 473.49 |
| KNOCKOFF-$k$NN | 19.87 | 19.42 | 19.81 | 19.56 | 409.18 |
| DDPI-RF | 16.10 | 16.68 | 17.85 | 16.88 | 634.85 |
| DDPI-$k$NN | 17.77 | 16.92 | 16.98 | 17.22 | 345.38 |
| PCA-RF | 18.72 | 20.06 | 22.18 | 20.32 | 424.30 |
| PCA-$k$NN | 16.53 | 16.31 | 20.83 | 17.89 | 326.75 |
| BNN-RF | 21.64 | 20.69 | 20.46 | 20.93 | 634.85 |
| BNN-$k$NN | 16.68 | 16.44 | 18.23 | 17.12 | 308.90 |
| VAEs-RF | 15.14 | 18.66 | 17.97 | 17.26 | 263.85 |
| VAEs-$k$NN | 4.60 | 8.50 | 7.83 | 6.98 | 443.85 |
| **PMFI-RF** | **22.75** | **22.73** | **27.04** | **24.17** | **218.18** |
| **PMFI-$k$NN** | 19.01 | 19.59 | 20.38 | 19.66 | 278.61 |

PMFI-$k$NN models. We observe the convergence of out-of-sample profit as the number of observed samples increases, aligning with our theoretical analysis in Theorem 4. We also observe that the out-of-sample performance closely approximates the underlying optimal level even with very small sample sizes. This suggests that our solution method demonstrates strong robustness in small-sample scenarios. Additionally, Figure 5 depicts the convergence of optimal order quantity decisions towards actual demand under current pricing decisions as the number of observations increases.

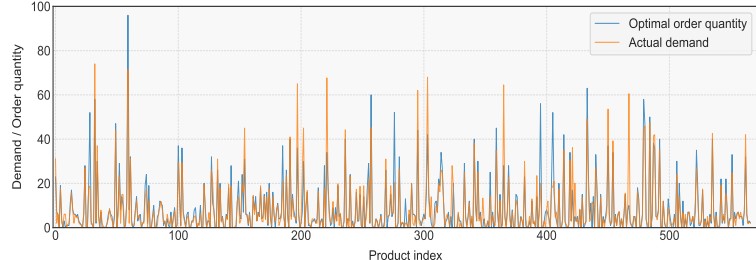

Figure 4: Comparison of resulting order quantities in PMFI-RF model vs. Actual demand

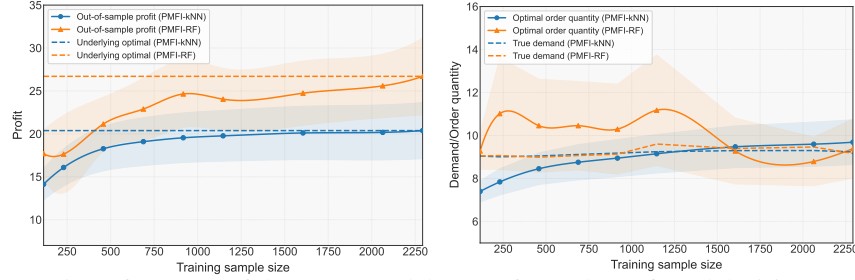

Figure 5: Performance of the PMFI-model: Out-of-sample profit and decision accuracy

## 6 CONCLUSION

This paper introduces a semi-parametric decision-making framework designed to alleviate the inevitable confounding effects present in contextual stochastic optimization problems characterized by endogeneity. Both theoretical and experimental results validate that the proposed methodology effectively mitigates the repercussions arising from the unmet assumption of unconfoundedness. Furthermore, our developed solution framework demonstrates superior performance in practical applications compared to conventional and state-of-the-art methods through synthetic and real datasets.

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
