APPENDIX

# A Proofs

## A.1 Proof of Section 2

*Proof of Theorem 1.* Based on Assumption 2, $\mathbb{E}\left[\pi\left(\boldsymbol{q};\mathbf{y}\right)\mid\boldsymbol{z},\boldsymbol{q}\right]$ is continuous in $\boldsymbol{z}$. For any given $\boldsymbol{z}$ and $\boldsymbol{q}$, $\breve{\mathbb{E}}\left[\pi\left(\boldsymbol{q};\mathbf{y}\right)\mid\boldsymbol{z},\boldsymbol{q}\right]=\sum_{i=1}^{n}w_i(\boldsymbol{z},\boldsymbol{q})\pi(\boldsymbol{q};\boldsymbol{y}_i)$, this function is not necessary continuous in $\boldsymbol{z}$, but we know that $\pi(\boldsymbol{q};\boldsymbol{y}_i)$ is bounded and the weight $w_i(\boldsymbol{z},\boldsymbol{q})$ is bounded by 1 for $\forall i\in[n]$. Therefore, we can obtain the conclusion that the supremum of $\left|\mathbb{E}\left[\pi\left(\boldsymbol{q};\mathbf{y}\right)\mid\boldsymbol{z},\boldsymbol{q}\right]-\breve{\mathbb{E}}\left[\pi\left(\boldsymbol{q};\mathbf{y}\right)\mid\boldsymbol{z},\boldsymbol{q}\right]\right|$ over $\boldsymbol{q}\in\mathbb{Q}$ exists. To obtain the final results, we need the following lemma.

**Lemma 1** *Suppose Assumption 2 holds, then we have*
$$\left|\mathbb{E}\left[\pi\left(\boldsymbol{q}^*;\mathbf{y}\right)\mid\boldsymbol{z},\boldsymbol{q}^*\right]-\mathbb{E}\left[\pi\left(\check{\boldsymbol{q}};\mathbf{y}\right)\mid\boldsymbol{z},\check{\boldsymbol{q}}\right]\right|\leq 2\sup_{\boldsymbol{q}\in\mathbb{Q}}\left|\mathbb{E}\left[\pi\left(\boldsymbol{q};\mathbf{y}\right)\mid\boldsymbol{z},\boldsymbol{q}\right]-\breve{\mathbb{E}}\left[\pi\left(\boldsymbol{q};\mathbf{y}\right)\mid\boldsymbol{z},\boldsymbol{q}\right]\right|.$$

*Proof of Lemma 1.* We know that
$$\boldsymbol{q}^*\in\arg\max_{\boldsymbol{q}\in\mathbb{Q}}\mathbb{E}\left[\pi\left(\boldsymbol{q};\mathbf{y}\right)\mid\boldsymbol{z},\boldsymbol{q}\right],$$
and
$$\check{\boldsymbol{q}}\in\arg\max_{\boldsymbol{q}\in\mathbb{Q}}\breve{\mathbb{E}}\left[\pi\left(\boldsymbol{q};\mathbf{y}\right)\mid\boldsymbol{z},\boldsymbol{q}\right].$$
Then we have
$$|\underbrace{\mathbb{E}\left[\pi\left(\boldsymbol{q}^*;\mathbf{y}\right)\mid\boldsymbol{z},\boldsymbol{q}^*\right]}_{(a)}-\underbrace{\breve{\mathbb{E}}\left[\pi\left(\check{\boldsymbol{q}};\mathbf{y}\right)\mid\boldsymbol{z},\check{\boldsymbol{q}}\right]}_{(b)}|$$
$$\leq\begin{cases}\left|\mathbb{E}\left[\pi\left(\check{\boldsymbol{q}};\mathbf{y}\right)\mid\boldsymbol{z},\check{\boldsymbol{q}}\right]-\breve{\mathbb{E}}\left[\pi\left(\check{\boldsymbol{q}};\mathbf{y}\right)\mid\boldsymbol{z},\check{\boldsymbol{q}}\right]\right| & \text{if }(a)\leq(b)\\\left|\mathbb{E}\left[\pi\left(\boldsymbol{q}^*;\mathbf{y}\right)\mid\boldsymbol{z},\boldsymbol{q}^*\right]-\breve{\mathbb{E}}\left[\pi\left(\boldsymbol{q}^*;\mathbf{y}\right)\mid\boldsymbol{z},\boldsymbol{q}^*\right]\right| & \text{if }(a)>(b)\end{cases}\tag{18}$$
$$\leq\sup_{\boldsymbol{q}\in\mathbb{Q}}\left|\mathbb{E}\left[\pi\left(\boldsymbol{q};\mathbf{y}\right)\mid\boldsymbol{z},\boldsymbol{q}\right]-\breve{\mathbb{E}}\left[\pi\left(\boldsymbol{q};\mathbf{y}\right)\mid\boldsymbol{z},\boldsymbol{q}\right]\right|,$$
thus we can obtain that
$$\left|\mathbb{E}\left[\pi\left(\boldsymbol{q}^*;\mathbf{y}\right)\mid\boldsymbol{z},\boldsymbol{q}^*\right]-\mathbb{E}\left[\pi\left(\check{\boldsymbol{q}};\mathbf{y}\right)\mid\boldsymbol{z},\check{\boldsymbol{q}}\right]\right|$$
$$\leq\left|\mathbb{E}\left[\pi\left(\boldsymbol{q}^*;\mathbf{y}\right)\mid\boldsymbol{z},\boldsymbol{q}^*\right]-\breve{\mathbb{E}}\left[\pi\left(\check{\boldsymbol{q}};\mathbf{y}\right)\mid\boldsymbol{z},\check{\boldsymbol{q}}\right]\right|+\left|\breve{\mathbb{E}}\left[\pi\left(\check{\boldsymbol{q}};\mathbf{y}\right)\mid\boldsymbol{z},\check{\boldsymbol{q}}\right]-\mathbb{E}\left[\pi\left(\check{\boldsymbol{q}};\mathbf{y}\right)\mid\boldsymbol{z},\check{\boldsymbol{q}}\right]\right|\tag{19}$$
$$\leq 2\sup_{\boldsymbol{q}\in\mathbb{Q}}\left|\mathbb{E}\left[\pi\left(\boldsymbol{q};\mathbf{y}\right)\mid\boldsymbol{z},\boldsymbol{q}\right]-\breve{\mathbb{E}}\left[\pi\left(\boldsymbol{q};\mathbf{y}\right)\mid\boldsymbol{z},\boldsymbol{q}\right]\right|,$$
where the second inequality follows from (A.1), that complete the proof of Lemma 1. $\qquad\square$

Consequently, we can obtain the following inequality:
$$\left|\mathbb{E}\left[\pi\left(\boldsymbol{q}^*;\mathbf{y}\right)\mid\boldsymbol{z},\boldsymbol{q}^*\right]-\mathbb{E}\left[\pi\left(\check{\boldsymbol{q}};\mathbf{y}\right)\mid\boldsymbol{z},\check{\boldsymbol{q}}\right]\right|$$
$$\leq 2\sup_{\boldsymbol{q}\in\mathbb{Q}}\left|\mathbb{E}\left[\pi\left(\boldsymbol{q};\mathbf{y}\right)\mid\boldsymbol{z},\boldsymbol{q}\right]-\breve{\mathbb{E}}\left[\pi\left(\boldsymbol{q};\mathbf{y}\right)\mid\boldsymbol{z},\boldsymbol{q}\right]\right|\tag{20}$$
$$\leq 2\sup_{\boldsymbol{q}_2\in\mathbb{Q}}\sup_{\boldsymbol{q}_1\in\mathbb{Q}}\left|\mathbb{E}\left[\pi\left(\boldsymbol{q}_2;\mathbf{y}\right)\mid\boldsymbol{z},\boldsymbol{q}_1\right]-\breve{\mathbb{E}}\left[\pi\left(\boldsymbol{q}_2;\mathbf{y}\right)\mid\boldsymbol{z},\boldsymbol{q}_1\right]\right|.$$
In the second inequality, we take the supremum over the decision $\boldsymbol{q}$ twice: $\boldsymbol{q}_1$ and $\boldsymbol{q}_2$. Here, $\boldsymbol{q}_1$ only affects the distribution of the uncertainty $\mathbf{y}$, and $\boldsymbol{q}_2$ influences the utility function $\pi$. The RHS of (20) is always an upper bound for the RHS of (19), as $\boldsymbol{q}_1=\boldsymbol{q}_2$ is a special case for this uniform gap.

**Lemma 2** *Suppose Assumption 1 and 2 holds, and the historical observations are i.i.d., then we have*
$$\mathbb{P}\left\{\sup_{(\boldsymbol{z},\boldsymbol{q}_1)\in\mathbb{A}}\left|\mathbb{E}\left[\pi\left(\boldsymbol{q}_2;\mathbf{y}\right)\mid\boldsymbol{z},\boldsymbol{q}_1\right]-\breve{\mathbb{E}}\left[\pi\left(\boldsymbol{q}_2;\mathbf{y}\right)\mid\boldsymbol{z},\boldsymbol{q}_1\right]\right|\geq\delta\right\}$$
$$\leq\left(\frac{4\sqrt{p+b}L_2 l}{\delta}\right)^{p+b}\exp\left\{-2ng^2\left(\frac{\delta}{4L_2}\right)^{2p+2b}\right\}\tag{21}$$
$$+2\left(\frac{25}{p+b}\right)\exp\left\{-\left(\frac{k\delta^2}{8\sigma_1^2}-2(p+b)\log n\right)\right\},$$

*for any $\delta \geq 2L_2 \left(\frac{k-1}{ng}\right)^{1/(p+b)}$ and $n > 2(p+b)$, where $l$ is a constant that depends on $\ell_p$-norm.*

**Lemma 3** *Suppose Assumption 1 and 2 holds, for any given $(z, q_1)$, $\sup_{q_1 \in \mathbb{Q}} \left| \mathbb{E}\left[\pi\left(q_2; \mathbf{y}\right) \mid z, q_1\right] - \check{\mathbb{E}}\left[\pi\left(q_2; \mathbf{y}\right) \mid z, q_1\right] \right|$ is $2L_1$-lipschtiz continuous in $q_2$.*

*Proof of Lemma 3:* For any $q_2, q_2' \in \mathbb{Q}$:

$$\left| \mathbb{E}\left[\pi\left(q_2; \mathbf{y}\right) \mid z, q_1\right] - \check{\mathbb{E}}\left[\pi\left(q_2; \mathbf{y}\right) \mid z, q_1\right] \right|$$
$$- \left| \mathbb{E}\left[\pi\left(q_2'; \mathbf{y}\right) \mid z, q_1\right] - \check{\mathbb{E}}\left[\pi\left(q_2'; \mathbf{y}\right) \mid z, q_1\right] \right|$$
$$\leq \left| \mathbb{E}[\pi(q_2; \mathbf{y}) - \pi(q_2'; \mathbf{y}) \mid z, q_1] \right| + \left| \check{\mathbb{E}}[\pi(q_2; \mathbf{y}) - \pi(q_2'; \mathbf{y}) \mid z, q] \right|$$
$$\leq \left| \mathbb{E}[L(\mathbf{y})|q_2 - q_2'| \mid z, q_1] \right| + \sum_{i=1}^{n} w_i(z, q)[\pi(q_2; y_i) - \pi(q_2'; y_i)] \qquad (22)$$
$$\leq L_1 |q_2 - q_2'| + \sum_{i=1}^{n} w_i(z, q)[\pi(q_2; y_i) - \pi(q_2'; y_i)]$$
$$\leq 2L_1 |q_2 - q_2'|,$$

where the third and fourth inequalities follow from the $L_1$ lipschtiz continuous of $\pi(q; Y)$ in $q$ for any $y \in \mathbb{Y}$ and the definition of $\check{\mathbb{E}}\left[\pi\left(q; \mathbf{y}\right) \mid z, q\right]$. Then we take the supremum over both sides of (22), we can obtain that

$$\sup_{q \in \mathbb{Q}} \left| \mathbb{E}\left[\pi\left(q_2; \mathbf{y}\right) \mid z, q_1\right] - \check{\mathbb{E}}\left[\pi\left(q_2; \mathbf{y}\right) \mid z, q_1\right] \right|$$
$$\leq \sup_{q \in \mathbb{Q}} \left| \mathbb{E}\left[\pi\left(q_2'; \mathbf{y}\right) \mid z, q_1\right] - \check{\mathbb{E}}\left[\pi\left(q_2'; \mathbf{y}\right) \mid z, q_1\right] \right| + 2L_1 |q_2 - q_2'|.$$

That completes the proof of Lemma 3. $\qquad\square$

**Lemma 4** *Suppose $\mathbb{Q}$ is compact with diameter $B$, then we have that*

$$\mathbb{P}\left\{ \sup_{q_2 \in \mathbb{Q}} \sup_{q_1 \in \mathbb{Q}} \left| \mathbb{E}\left[\pi\left(q_2; \mathbf{y}\right) \mid z, q_1\right] - \check{\mathbb{E}}\left[\pi\left(q_2; \mathbf{y}\right) \mid z, q_1\right] \right| > \delta \right\}$$
$$\leq \left(\frac{4lBL_1}{\delta}\right)^p \sup_{q_2 \in \mathbb{Q}} \mathbb{P}\left\{ \sup_{q_1 \in \mathbb{Q}} \left| \mathbb{E}\left[\pi\left(q_2; \mathbf{y}\right) \mid z, q_1\right] - \check{\mathbb{E}}\left[\pi\left(q_2; \mathbf{y}\right) \mid z, q_1\right] \right| > \frac{\delta}{2} \right\}.$$

The proof of Lemma 4 is same as Lemma 10 in Bertsimas & McCord (2019). Based on the above results, we have

$$\mathbb{P}\left\{ \left| \mathbb{E}\left[\pi\left(q^*; \mathbf{y}\right) \mid z, q^*\right] - \mathbb{E}\left[\pi\left(\check{q}; \mathbf{y}\right) \mid z, \check{q}\right] \right| > \delta \right\}$$
$$\leq \mathbb{P}\left\{ 2 \sup_{q_2 \in \mathbb{Q}} \sup_{q_1 \in \mathbb{Q}} \left| \mathbb{E}\left[\pi\left(q_2; \mathbf{y}\right) \mid z, q_1\right] - \check{\mathbb{E}}\left[\pi\left(q_2; \mathbf{y}\right) \mid z, q_1\right] \right| > \delta \right\}$$
$$\leq \left(\frac{8lBL_1}{\delta}\right)^p \sup_{q_2 \in \mathbb{Q}} \mathbb{P}\left\{ \sup_{q_1 \in \mathbb{Q}} \left| \mathbb{E}\left[\pi\left(q_2; \mathbf{y}\right) \mid z, q_1\right] - \check{\mathbb{E}}\left[\pi\left(q_2; \mathbf{y}\right) \mid z, q_1\right] \right| > \frac{\delta}{4} \right\} \qquad (23)$$
$$\leq \left(\frac{8lBL_1}{\delta}\right)^p \left(\frac{16\sqrt{p+b}L_2 l}{\delta}\right)^{p+b} \exp\left\{ -2ng^2 \left(\frac{\delta}{16L_2}\right)^{2p+2b} \right\}$$
$$+ 2\left(\frac{8lBL_1}{\delta}\right)^p \left(\frac{25}{p+b}\right) \exp\left\{ -\left(\frac{k\delta^2}{128\sigma^2} - 2(p+b)\log n\right) \right\},$$

where the first inequality follows from (20), the second inequality follows from Lemma 4, and the last inequality follows from Lemma 2. From this results we can deduce that to satisfy the RHS of (23) smaller or equal to $\alpha$, we need that

$$\delta \geq \left(n2g^2\right)^{-\frac{1}{2p+2b}} 16L_2 \left(\frac{2p+b}{2p+2b} \log n + C_2\right)^{\frac{1}{2p+2b}}$$

$$\delta \geq 8\sqrt{2}\sigma n^{-\frac{\gamma}{2}} \left( \sqrt{\frac{4p + 4b + \gamma p}{2} \log n} + C_4 \right),$$

we set $C_1 \triangleq 16L_2 \left(2g^2\right)^{-\frac{1}{2p+2b}}$, $C_2 = \log \frac{2}{\delta} + \log 8lBL_1 \left(16\sqrt{b+p}L_2l\right)^{b+p} + \log \left( \frac{\left(2g^2\right)^{\frac{1}{2p+2b}}}{16L_2 \log \frac{2}{\delta}} \right)^{2p+b}$, $C_3 = 8\sqrt{2}\sigma$, $C_4 = \sqrt{\log \frac{4}{\alpha}} + \sqrt{\log \frac{25(8lBL_1)^p}{p+b}} + \sqrt{p \left( \log \frac{1}{8\sqrt{2}\sigma\sqrt{\log \frac{4}{\alpha}}} \right)^+}$,

That completes the proof of Theorem 1. $\qquad\square$

*Proof of Theorem 2.* We know that both $\mathbb{E}\left[\pi\left(\boldsymbol{q};\mathbf{y}\right) \mid \boldsymbol{x}, \boldsymbol{q}\right]$ and $\mathbb{E}\left[\pi\left(\boldsymbol{q};\mathbf{y}\right) \mid \boldsymbol{z}, \boldsymbol{q}\right]$ are continuous in $\boldsymbol{q}$, $\boldsymbol{q}^* \in \arg\max_{\boldsymbol{q}} \mathbb{E}\left[\pi\left(\boldsymbol{q};\mathbf{y}\right) \mid \boldsymbol{z}, \boldsymbol{q}\right]$ and $\boldsymbol{q}_\star \in \arg\max_{\boldsymbol{q}} \mathbb{E}\left[\pi\left(\boldsymbol{q};\mathbf{y}\right) \mid \boldsymbol{x}, \boldsymbol{q}\right]$. Then we have that

$$\begin{aligned}
&\mathbb{E}\left[\pi\left(\boldsymbol{q}_\star;\mathbf{y}\right) \mid \boldsymbol{x}, \boldsymbol{q}_\star\right] - \mathbb{E}\left[\pi\left(\boldsymbol{q}^*;\mathbf{y}\right) \mid \boldsymbol{z}, \boldsymbol{q}^*\right] \\
&\leq \sup_{\boldsymbol{q}\in\mathbb{Q}} \left|\mathbb{E}\left[\pi\left(\boldsymbol{q};\mathbf{y}\right) \mid \boldsymbol{x}, \boldsymbol{q}\right] - \mathbb{E}\left[\pi\left(\boldsymbol{q};\mathbf{y}\right) \mid \boldsymbol{z}, \boldsymbol{q}\right]\right|.
\end{aligned} \tag{24}$$

Moreover, we have

$$\begin{aligned}
\mathcal{R}_2(\boldsymbol{x}, \boldsymbol{z}) &= \mathbb{E}\left[\pi\left(\boldsymbol{q}_\star;\mathbf{y}\right) \mid \boldsymbol{x}, \boldsymbol{q}_\star\right] - \mathbb{E}\left[\pi\left(\check{\boldsymbol{q}};\mathbf{y}\right) \mid \boldsymbol{x}, \check{\boldsymbol{q}}\right] \\
&= \mathbb{E}\left[\pi\left(\boldsymbol{q}^*;\mathbf{y}\right) \mid \boldsymbol{z}, \boldsymbol{q}^*\right] - \mathbb{E}\left[\pi\left(\check{\boldsymbol{q}};\mathbf{y}\right) \mid \boldsymbol{z}, \check{\boldsymbol{q}}\right] \\
&\quad + \mathbb{E}\left[\pi\left(\boldsymbol{q}_\star;\mathbf{y}\right) \mid \boldsymbol{x}, \boldsymbol{q}_\star\right] - \mathbb{E}\left[\pi\left(\boldsymbol{q}^*;\mathbf{y}\right) \mid \boldsymbol{z}, \boldsymbol{q}^*\right] \\
&\quad + \mathbb{E}\left[\pi\left(\check{\boldsymbol{q}};\mathbf{y}\right) \mid \boldsymbol{z}, \check{\boldsymbol{q}}\right] - \mathbb{E}\left[\pi\left(\check{\boldsymbol{q}};\mathbf{y}\right) \mid \boldsymbol{x}, \check{\boldsymbol{q}}\right] \\
&= \mathcal{R}_1(\boldsymbol{z}) + \mathbb{E}\left[\pi\left(\boldsymbol{q}_\star;\mathbf{y}\right) \mid \boldsymbol{x}, \boldsymbol{q}_\star\right] - \mathbb{E}\left[\pi\left(\boldsymbol{q}^*;\mathbf{y}\right) \mid \boldsymbol{z}, \boldsymbol{q}^*\right] \\
&\quad + \mathbb{E}\left[\pi\left(\check{\boldsymbol{q}};\mathbf{y}\right) \mid \boldsymbol{z}, \check{\boldsymbol{q}}\right] - \mathbb{E}\left[\pi\left(\check{\boldsymbol{q}};\mathbf{y}\right) \mid \boldsymbol{x}, \check{\boldsymbol{q}}\right] \\
&\leq \mathcal{R}_1(\boldsymbol{z}) + \sup_{\boldsymbol{q}} \left|\mathbb{E}\left[\pi\left(\boldsymbol{q};\mathbf{y}\right) \mid \boldsymbol{x}, \boldsymbol{q}\right] - \mathbb{E}\left[\pi\left(\boldsymbol{q};\mathbf{y}\right) \mid \boldsymbol{z}, \boldsymbol{q}\right]\right| \\
&\quad + \sup_{\boldsymbol{q}} \left|\mathbb{E}\left[\pi\left(\boldsymbol{q};\mathbf{y}\right) \mid \boldsymbol{x}, \boldsymbol{q}\right] - \mathbb{E}\left[\pi\left(\boldsymbol{q};\mathbf{y}\right) \mid \boldsymbol{z}, \boldsymbol{q}\right]\right| \\
&= \mathcal{R}_1(\boldsymbol{z}) + 2\sup_{\boldsymbol{q}} \left|\mathbb{E}\left[\pi\left(\boldsymbol{q};\mathbf{y}\right) \mid \boldsymbol{x}, \boldsymbol{q}\right] - \mathbb{E}\left[\pi\left(\boldsymbol{q};\mathbf{y}\right) \mid \boldsymbol{z}, \boldsymbol{q}\right]\right|,
\end{aligned}$$

where the inequality follows from (24). If we further have $b = m$, then $\boldsymbol{x}$ and $\boldsymbol{z}$ have the same dimension, based on the Lipschitz continuous, we have that

$$\begin{aligned}
\mathcal{R}_2(x, z) &\leq \mathcal{R}_1(\boldsymbol{z}) + 2\sup_{\boldsymbol{q}} \left|\mathbb{E}\left[\pi\left(\boldsymbol{q};\mathbf{y}\right) \mid \boldsymbol{x}, \boldsymbol{q}\right] - \mathbb{E}\left[\pi\left(\boldsymbol{q};\mathbf{y}\right) \mid \boldsymbol{z}, \boldsymbol{q}\right]\right| \\
&\leq \mathcal{R}_1(\boldsymbol{z}) + 2L_2 \left\|\boldsymbol{x} - \boldsymbol{z}\right\|.
\end{aligned}$$

That completes the proof. $\qquad\square$

### A.2 PROOF OF SECTION 4

*Proof of Theorem 3.* Following the same analysis in the proof of Theorem 2 , we have

$$\begin{aligned}
\mathcal{R}_3(\boldsymbol{x}, \hat{\boldsymbol{x}}) &= \mathbb{E}\left[\pi\left(\boldsymbol{q}_\star;\mathbf{y}\right) \mid \boldsymbol{x}, \boldsymbol{q}_\star\right] - \mathbb{E}\left[\pi\left(\hat{\boldsymbol{q}};\mathbf{y}\right) \mid \boldsymbol{x}, \hat{\boldsymbol{q}}\right] \\
&= \mathbb{E}\left[\pi\left(\boldsymbol{q}^\sharp;\mathbf{y}\right) \mid \hat{\boldsymbol{x}}, \boldsymbol{q}^\sharp\right] - \mathbb{E}\left[\pi\left(\hat{\boldsymbol{q}};\mathbf{y}\right) \mid \hat{\boldsymbol{x}}, \hat{\boldsymbol{q}}\right] \\
&\quad + \mathbb{E}\left[\pi\left(\boldsymbol{q}_\star;\mathbf{y}\right) \mid \boldsymbol{x}, \boldsymbol{q}_\star\right] - \mathbb{E}\left[\pi\left(\boldsymbol{q}^\sharp;\mathbf{y}\right) \mid \hat{\boldsymbol{x}}, \boldsymbol{q}^\sharp\right] \\
&\quad + \mathbb{E}\left[\pi\left(\hat{\boldsymbol{q}};\mathbf{y}\right) \mid \hat{\boldsymbol{x}}, \hat{\boldsymbol{q}}\right] - \mathbb{E}\left[\pi\left(\hat{\boldsymbol{q}};\mathbf{y}\right) \mid \boldsymbol{x}, \hat{\boldsymbol{q}}\right] \\
&= \mathcal{R}_1(\hat{\boldsymbol{x}}) + \mathbb{E}\left[\pi\left(\boldsymbol{q}_\star;\mathbf{y}\right) \mid \boldsymbol{x}, \boldsymbol{q}_\star\right] - \mathbb{E}\left[\pi\left(\hat{\boldsymbol{q}};\mathbf{y}\right) \mid \hat{\boldsymbol{x}}, \hat{\boldsymbol{q}}\right] \\
&\quad + \mathbb{E}\left[\pi\left(\hat{\boldsymbol{q}};\mathbf{y}\right) \mid \hat{\boldsymbol{x}}, \hat{\boldsymbol{q}}\right] - \mathbb{E}\left[\pi\left(\hat{\boldsymbol{q}};\mathbf{y}\right) \mid \boldsymbol{x}, \hat{\boldsymbol{q}}\right] \\
&\leq \mathcal{R}_1(\hat{\boldsymbol{x}}) + \sup_{\boldsymbol{q}} \left|\mathbb{E}\left[\pi\left(\boldsymbol{q};\mathbf{y}\right) \mid \boldsymbol{x}, \boldsymbol{q}\right] - \mathbb{E}\left[\pi\left(\boldsymbol{q};\mathbf{y}\right) \mid \hat{\boldsymbol{x}}, \boldsymbol{q}\right]\right| \\
&\quad + \sup_{\boldsymbol{q}} \left|\mathbb{E}\left[\pi\left(\boldsymbol{q};\mathbf{y}\right) \mid \boldsymbol{x}, \boldsymbol{q}\right] - \mathbb{E}\left[\pi\left(\boldsymbol{q};\mathbf{y}\right) \mid \hat{\boldsymbol{x}}, \boldsymbol{q}\right]\right| \\
&= \mathcal{R}_1(\hat{\boldsymbol{x}}) + 2\sup_{\boldsymbol{q}} \left|\mathbb{E}\left[\pi\left(\boldsymbol{q};\mathbf{y}\right) \mid \boldsymbol{x}, \boldsymbol{q}\right] - \mathbb{E}\left[\pi\left(\boldsymbol{q};\mathbf{y}\right) \mid \hat{\boldsymbol{x}}, \boldsymbol{q}\right]\right| \\
&\leq \mathcal{R}_1(\hat{\boldsymbol{x}}) + 2L_2 \|\boldsymbol{x} - \hat{\boldsymbol{x}}\|.
\end{aligned}$$

As for the second part, we have that with probability at least $1 - c_2 \exp\{-c_3 \log m\}$:

$$\|\hat{\boldsymbol{x}} - \boldsymbol{x}\|^2$$

$$\leq c_1' \frac{\|\boldsymbol{X} - \hat{\boldsymbol{X}}\|_F^2}{n+1}$$

$$= c_1' \frac{\|\boldsymbol{A} - \hat{\boldsymbol{A}}\|_F^2}{n+1}$$

$$\leq c_1' c_2' \frac{\alpha_{sp}^2(\boldsymbol{A})\sigma_2^2}{K^2 e^{-4\eta\|\boldsymbol{A}\|_{\max}}} \frac{rm\log m}{|\Omega|}\|\boldsymbol{A}\|_F^2 m$$

$$\leq c_1' c_2' \frac{\alpha_{sp}^2(\boldsymbol{A})\sigma_2^2}{K^2 e^{-4\eta\|\boldsymbol{A}\|_{\max}}} \frac{rm^2\log m}{|\Omega|}\|\boldsymbol{A}\|_F^2,$$

where the first inequality follows from Assumption 4, the first equality follows from that the loading matrix $\boldsymbol{V}$ is an identity matrix, the second inequality follows from corollary 1 in Gunasekar et al. (2014). Consequently, we have that with probability at least $1 - c_2 \exp\{-c_3 \log m\}$:

$$2L_2\|\hat{\boldsymbol{x}} - \boldsymbol{x}\|$$

$$\leq 2L_2\sqrt{c_1'c_2'}\frac{\alpha_{sp}(\boldsymbol{A})\sigma_2}{Ke^{-2\eta\|\boldsymbol{A}\|_{\max}}}\sqrt{\frac{rm^2\log m}{|\Omega|}}\|\boldsymbol{A}\|_F$$

$$\leq 2L_2\sqrt{c_1'c_2'}\frac{\alpha_{sp}(\boldsymbol{A})\sigma_2\sigma_{\max}(\boldsymbol{A})}{Ke^{-2\eta\|\boldsymbol{A}\|_{\max}}}\sqrt{\frac{r^3m^2\log m}{|\Omega|}},$$

where the inequality follows from that $\|\cdot\|_F \leq r\|\cdot\|$. We set $c_4 \triangleq \frac{2L_2\sqrt{c_1'c_2'}}{K}$ and $\zeta_{m,r} \triangleq \frac{c_4\alpha_{sp}(\boldsymbol{A})\sigma_{\max}(\boldsymbol{A})\sigma_2}{e^{-2\eta\|\boldsymbol{A}\|_{\max}}}\sqrt{\frac{r^3m^2\log m}{|\Omega|}}$ for simplicity. As for the first part $\mathcal{R}_1(\hat{\boldsymbol{x}})$, we can draw the conclusion that with probability at least $1 - \alpha$ where $0 < \alpha < 1$, we have that $\mathcal{R}_1(\hat{\boldsymbol{x}}) \leq \xi_{\alpha,p,m}$. Therefore, we have

$$\mathbb{P}\{\mathcal{R}_3(\boldsymbol{x}, \hat{\boldsymbol{x}}) > \xi_{\alpha,p,m} + \zeta_{m,r}\}$$
$$\leq \mathbb{P}\{\mathcal{R}_1(\hat{\boldsymbol{x}}) > \xi_{\alpha,p,m}\} + \mathbb{P}\{2L_2\|\boldsymbol{x} - \hat{\boldsymbol{x}}\| > \zeta_{m,r}\}$$
$$\leq \alpha + c_3 \exp\{-c_4 \log m\}.$$

Thus we have that with probability at least $1 - \alpha - c_3\exp\{-c_4\log m\}$, $\mathcal{R}_3(\boldsymbol{x}, \hat{\boldsymbol{x}}) \leq \xi_{\alpha,p,m} + \zeta_{m,r}$. That completes the proof. $\square$

*Proof of Theorem 4.* From the proof of Theorem 4, we know that $\mathcal{R}_3(x, \hat{\boldsymbol{x}}) \leq \mathcal{R}_1(\hat{\boldsymbol{x}}) + 2L_2\|\boldsymbol{x} - \hat{\boldsymbol{x}}\|$. As for the first part $\mathcal{R}_1(\hat{\boldsymbol{x}})$, we can draw the conclusion that with probability at least $1 - \alpha$ where $0 < \alpha < 1$, we have that $\mathcal{R}_1(\hat{\boldsymbol{x}}) \leq \xi_{\alpha,p,m}$. For the second part, we have

$$2L_2\|\boldsymbol{x} - \hat{\boldsymbol{x}}\| = 2L_2\sqrt{\|\boldsymbol{x} - \hat{\boldsymbol{x}}\|^2}$$

$$\leq 2L_2\sqrt{c_1'\frac{\inf_{\boldsymbol{B}\in\mathbb{O}_r}\|\hat{\boldsymbol{X}} - \boldsymbol{X}\boldsymbol{B}\|_F^2}{n+1}}$$

$$\leq 2L_2\sqrt{c_1'\frac{2\|\sin\angle(\boldsymbol{X}, \hat{\boldsymbol{X}})\|_F^2}{n+1}}$$

$$\leq 2L_2\sqrt{c_1'\frac{2r^2\|\sin\angle(\boldsymbol{X}, \hat{\boldsymbol{X}})\|^2}{n+1}}$$

$$= \sqrt{8c_1'}L_2 r\frac{\|\sin\angle(\boldsymbol{X}, \hat{\boldsymbol{X}})\|}{\sqrt{n+1}}$$

$$\leq \sqrt{8c_1'}L_2 r\frac{\frac{\|\hat{\boldsymbol{A}} - \boldsymbol{A}\|_F}{\sigma_{\min}(\hat{\boldsymbol{A}})}}{\sqrt{n+1}}$$

$$\leq \sqrt{8c_1'}L_2 r \frac{\sqrt{n+1}\sqrt{c_1'c_2'}\frac{c_4\alpha_{sp}(\boldsymbol{A})\sigma_2\sigma_{\max}(\boldsymbol{A})}{e^{-2\eta\|\boldsymbol{A}\|_{\max}}}\sqrt{\frac{r^3 b^2 \log b}{|\Omega|}}}{\sqrt{n+1}\sigma_{\min}(\hat{\boldsymbol{A}})}$$

$$= \frac{\sqrt{8c_1'c_1'c_2'}L_2\frac{c_4\alpha_{sp}(\boldsymbol{A})\sigma_2\sigma_{\max}(\boldsymbol{A})}{e^{-2\eta\|\boldsymbol{A}\|_{\max}}}\sqrt{\frac{r^5 b^2 \log b}{|\Omega|}}}{\sigma_{\min}(\hat{\boldsymbol{A}})},$$

where the first inequality follows from Assumption 4, the second inequality follows from the property of $\sin\angle$ such that $\inf_{\boldsymbol{B}\in\mathbb{O}_r}\|\hat{\boldsymbol{X}}-\boldsymbol{X}\boldsymbol{B}\|_F \leq \sqrt{2}\|\sin\angle(\boldsymbol{X},\hat{\boldsymbol{X}})\|_F$, the fourth inequality follows from the Wedin's $\sin\angle$ theorem, and the last inequality follows from corollary 1 in Gunasekar et al. (2014). We set $c_5 \triangleq \sqrt{8c_1'c_1'c_2'}L_2c_4$, the RHS of (25) can be simplified as $\frac{c_5\zeta_{b,r}}{\sigma_{\min}(\hat{\boldsymbol{A}})}$. Note that the estimated rank $\hat{r}$ is often large enough to ensure the equivalence of (11) and (12), thus the smallest singular value of $\hat{\boldsymbol{A}}$ is bounded above 0. By setting $\varphi_{b,r} \triangleq \frac{c_5 r\zeta_{b,r}}{\sigma_{\min}(\hat{\boldsymbol{A}})}$, we complete the proof. □

## B  PROBLEM EXAMPLES

The proposed decision framework in this paper applies to a variety of optimization problems. We present two examples in the following.

***Example 1. Joint inventory and pricing control***. In this problem, uncertainty y represents product demand, and the decision vector is $\boldsymbol{q} = (q_1, q_2)$ where $q_1$ is the price, which influences the uncertainty, and $q_2$ is the inventory, which is independent of the demand. The utility function is $\pi(\boldsymbol{q}; y) = (q_1-c)q_2 - q_1(q_2-y)^+$ where $c$ is the unit cost. The data size is then $\{\boldsymbol{z}_i, q_{1i}, y_i\}_{i=1,\dots,n}$. The PI-model for this problem is:

$$\max_{\boldsymbol{q}\in\mathbb{Q}} \mathbb{E}\left[(q_1-c)q_2 - q_1(q_2-y(q_1))^+ \mid \boldsymbol{z}\right],$$

then the PMFI-model can be formulated as:

$$\max_{q_1\in\mathbb{Q}_1, q_2\in\mathbb{Q}_2} \hat{\mathbb{E}}\left[(q_1-c)q_2 - q_1(q_2-y)^+ \mid \hat{\boldsymbol{x}}, q_1\right] = \sum_{i=1}^{n} w_i(\hat{\boldsymbol{x}}, q_1)[(q_1-c)q_2 - q_1(q_2-y_i)^+].$$

***Example 2. Advertising and personalized interest rates optimization***. In this problem, the uncertainty y is the indicator for whether or not the consumer applied for the loan. The decision vector is $\boldsymbol{q} = (q_1, q_2)$ where $q_1$ denotes the advertising content and $q_2$ is the interest rate offered. The historical dataset is then $\{\boldsymbol{z}_i, \boldsymbol{q}_i, y_i\}_{i=1,\dots,n}$. The PI-model for this problem is:

$$\max_{\boldsymbol{q}\in\mathbb{Q}} \mathbb{E}\left[q_2 \mathbf{1}_{y(q_1)=1}\right],$$

then the PMFI-mdoel can be formulated as:

$$\max_{(q_1,q_2)\in\mathbb{Q}} \hat{\mathbb{E}}\left[q_2 \mathbf{1}_{y=1} \mid \hat{\boldsymbol{x}}, \boldsymbol{q}\right] = \sum_{i=1}^{n} w_i(\hat{\boldsymbol{x}}, \boldsymbol{q})[q_2 \cdot \mathbf{1}_{y_i=1}].$$

***Example 3. Personalized pricing problem***. The DMs decide a price $q$ for customer with random valuation v. Then the uncertainty y is whether the customer will purchase the product, i.e. $y \triangleq \mathbf{1}_{q\leq v}$. The historical dataset is then $\{\boldsymbol{z}_i, q_i, y_i\}_{i=1,\dots,n}$. The PI-model for this problem is:

$$\max_{\boldsymbol{q}\in\mathbb{Q}} \mathbb{E}\left[q \cdot y(q)\right],$$

then the PMFI-mdoel can be formulated as:

$$\max_{q\in\mathbb{Q}} \hat{\mathbb{E}}\left[q_2 \mathbf{1}_{y=1} \mid \hat{\boldsymbol{x}}, q\right] = \sum_{i=1}^{n} w_i(\hat{\boldsymbol{x}}, q)q y_i.$$

It is important to note that the above examples are merely illustrative; our proposed decision framework is applicable to all decision problems where confounding effects may exist. By estimating the confounders matrix using matrix completion methods, we believe this process can be integrated with other data-driven decision-making approaches. Additionally, our framework can also be applied in scenarios where decisions do not affect uncertainty.

# C INFORMATION ON DATASET

## C.1 SYNTHETIC DATA

We set $\mathbb{Q}_1 \triangleq \{2.5, 3.0, 3.5, 4.0, 4.5, 5.0, 5.5, 6.0\}$, $m = 5$, and $\boldsymbol{X} \in \mathbb{R}^{(n+1) \times 5}$ to be the confounders matrix, where $X_{i,1} \sim \mathcal{U}[-1, 1]$, $X_{i,2}, X_{i,3} \sim \mathcal{N}(0, 1)$, and $X_{i,4}, X_{i,5} \sim \mathcal{U}[0, 1]$ for $i = 1, ..., n$. Let $\boldsymbol{a} = (1, 2, 2, 2, 2)^\top$, and $t_i \triangleq \boldsymbol{X}_{i,:}\boldsymbol{a}$ then the historical price is $q_{1i} = 2.5$ if $t_i \in [-9, -6)$, $q_{1i} = 3.0$ if $t_i \in [-6, -3)$, $q_{1i} = 3.5$ if $t_i \in [-3, 0)$, $q_{1i} = 4.0$ if $t_i \in [0, 2)$, $q_{1i} = 4.5$ if $t_i \in [2, 4)$, $q_{1i} = 5.0$ if $t_i \in [4, 6)$, $q_{1i} = 5.5$ if $t_i \in [6, 9)$, $q_{1i} = 6.0$ if $t_i \in [9, 12)$. The historical demand is generated by

$$y_i = \max\left\{0, b_0 + b_1 \cdot q_{1i} + b_2 \cdot x_{i1} + x_{i2}^2 + x_{i3} + x_{i4} + x_{i5} \cdot \epsilon\right\},$$

where $b_0 = 300$, $b_1 = -35$, $b_2 = 200$, and $\epsilon \sim \mathcal{N}(0, 1)$. The dimension of proxy features is set to be $b = \{5, 10, 50, 100, 500, 800\}$, each entry $V_{k,j}$ for $k \in [5]$ and $j \in [b]$ in the loading matrix $\boldsymbol{V} \in \mathbb{R}^{5 \times b}$ is generated by the normal distribution $\mathcal{N}(0, 1)$. Let $\boldsymbol{A} \triangleq \boldsymbol{X}\boldsymbol{V}$, then with probability 0.1, the proxy feature $Z_{i,j}$ for $i \in [n + 1]$ and $k \in [b]$ is missing, i.e., $Z_{i,j} =$ NaN, and with probability 0.9, the proxy feature $Z_{i,j}$ is generated by $z_{ij} \sim \mathcal{N}(A_{i,j}, 5)$.

## C.2 REAL-WORLD DATA

Our real dataset is sourced from our collaborative partner, AEON, which is one of the largest retail companies in Japan. Our original dataset consists of a total of 171,709 sales records from four stores, covering the period from August 1st to August 31st, 2023. The original dataset includes 712 features, such as product category, weather-related, sales-related, price-related, discount-related features, among others. For the current experiment, we have selected the sales data of products with prices ranging from 0 to 5 from one store for the week of August 1st to August 7th for preprocessing and experimentation.

The first dataset comprises 2,866 products, each characterized by sales volume, price, and 438 proxy features. We employ an 80-20 train-test split, with 2,293 products for training and 573 ones for testing. We filter products with prices in the range of $[4, 6]$. We observe that product prices cluster around the values of 4, 4.5, 5, 5.5, and 6. Consequently, we assume that the feasible domain for prices is $\mathbb{Q} = \{4, 4.5, 5, 5.5, 6\}$. In the test set, we utilize the random forest method to construct demand at different price levels. Table 2 provides an example of the dataset with partial proxy features.

Table 2: Sample of dataset with partial proxy features

| demand | price | section | article | class | merch_type | cost_in_sell_unit | weather | wind_speed | ... |
|--------|-------|---------|---------|-------|------------|-------------------|---------|------------|-----|
| 16 | 6 | 134 | 344 | 34401 | 1 | 2.39 | 1 | 2 | ... |

Figure 6 displays the optimal order quantities and corresponding actual demands under the PMFI-model when using a $k$NN weight function. Although the decisions are slightly inferior to those made under the PMFI-model with an RF weight function, this method still provides accurate demand predictions and results in good decision-making.

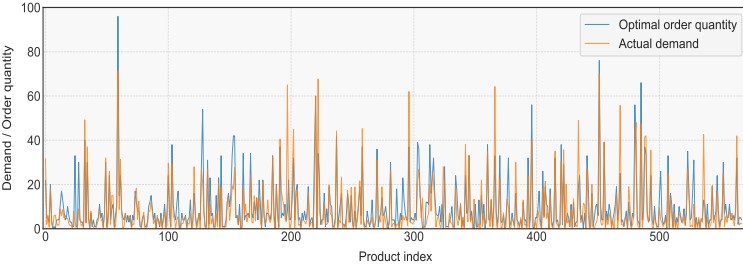

Figure 6: Comparison of optimal order quantities in PMFI-$k$NN model vs. Actual demand

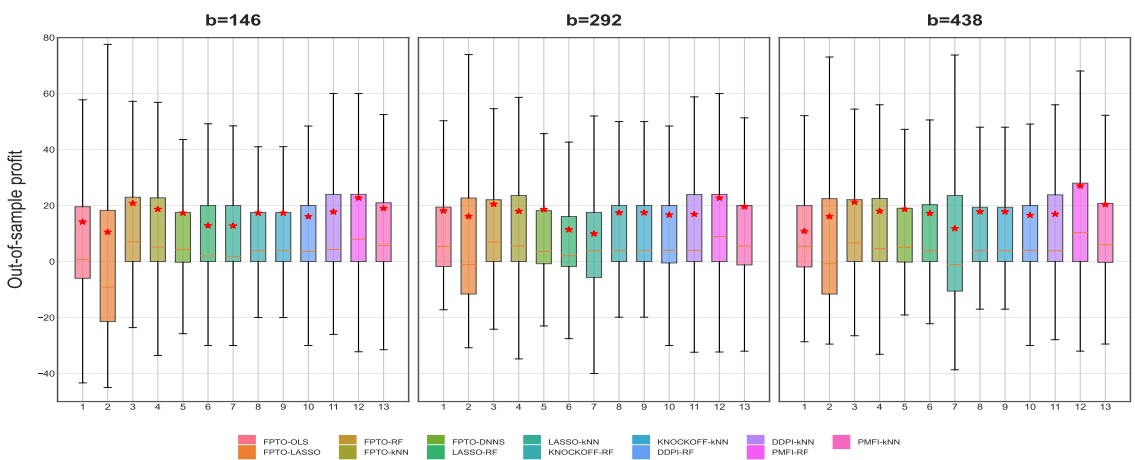

Figure 7: Out-of-sample profit across varying proxy feature dimensions and methods

Figure 7 shows the out-of-sample performance of various methods across different feature dimensions. We find that our proposed method achieves the highest out-of-sample average profit while maintaining a smaller profit variance.

Figure 8 illustrates the in-sample performance of different methods. While the FSTO framework and DDPI models exhibit strong in-sample performance, their out-of-sample performance is poor, indicating an issue with overfitting. Our model addresses this overfitting problem by inferring confounders

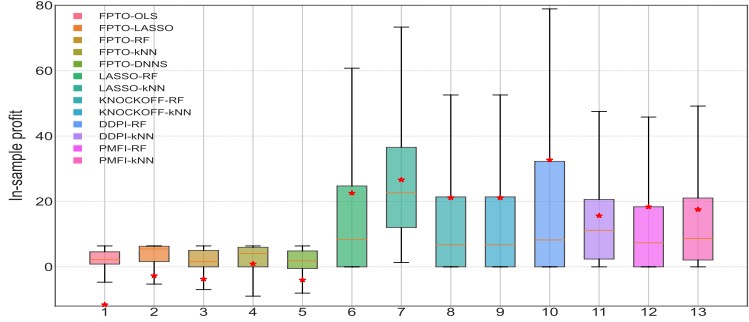

Figure 8: In-sample profit across varying methods (b=438)

We also conducted experiments using data from another store during the period of August 8th to August 14th. The second dataset comprises 2,874 products, each characterized by sales volume, price, and 438 proxy features. We employ an 80-20 train-test split, with 2,299 products for training and 575 ones for testing.

Table 3 compares the performance of the proposed method against other methods on a new dataset. The results are consistent with the findings presented in the main text, where our method demonstrates superior performance. Figure 9 illustrates the optimal order quantities and corresponding actual demands under the PMFI-RF model, indicating that our decision framework maintains high decision accuracy across different datasets. Therefore, the proposed method not only performs well relative to other methods across various datasets but also exhibits high decision accuracy, thereby confirming its robustness.

Table 3: Out-of-sample performance comparison on the second dataset

| Method | Out-of-sample profit | Decision accuracy (MSE) |
|---|---|---|
| FPTO-OLS | 17.38 | 773.65 |
| FPTO-LASSO | 19.64 | 515.62 |
| FPTO-RF | 26.16 | 478.86 |
| FPTO-$k$NN | 20.43 | 381.42 |
| FPTO-DNNs | 25.05 | 471.31 |
| LASSO-RF | 25.12 | 432.69 |
| LASSO-$k$NN | 24.44 | 358.38 |
| KNOCKOFF-RF | 22.88 | 471.75 |
| KNOCKOFF-$k$NN | 21.73 | 369.10 |
| DDPI-RF | 23.05 | 654.57 |
| DDPI-$k$NN | 19.52 | 401.41 |
| PCA-RF | 27.19 | 341.53 |
| PCA-$k$NN | 21.48 | 362.97 |
| BNN-RF | 25.00 | 415.61 |
| BNN-$k$NN | 19.17 | 402.16 |
| VAEs-RF | 19.97 | 370.06 |
| VAEs-$k$NN | 10.37 | 500.98 |
| **PMFI-RF** | **33.74** | **168.60** |
| **PMFI-$k$NN** | 24.70 | 318.35 |

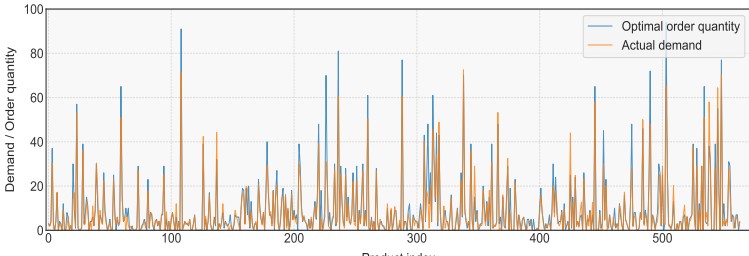

Figure 9: Comparison of optimal order quantities in PMFI-RF model vs. Actual demand on the second dataset

# D    DETAILED EXAMPLE

**Confounders**: In pricing decisions, features such as product quality, design style, and brand reputation concurrently influence both the pricing decision and the unknown demand for the product.

**Proxy of confounders**: In pricing problems, confounders such as weather conditions can be easily observed, whereas confounders like brand image are more challenging to quantify. DMs might use features such as customer ratings as proxies for brand image.

**Unconfoundedness**: In the context of pricing problems, Assumption 1 implies that the proxy features include all factors that simultaneously affect both pricing and demand. Therefore, conditional on the proxy features $z$, the confounding between price decisions and demand is isolated, they can be considered as if randomly generated.

To provide more intuition, suppose one is interested in the effect of a treatment on a certain disease. In this case, the uncertainty $\mathbf{y}$ represents whether a patient recovers from the disease, $\mathbf{q}$ represents whether a patient is prescribed the treatment, and $\mathbf{z}$ represents the covariate variables of a patient such as age and gender. Assumption 1 then requires that the data satisfies the following: conditioned on the patient's covariate information, whether to prescribe the treatment for a patient must be independent of the hypothetical recovery outcome.