# OpenReview forum: "A Proxy Matrix-based Framework for Contextual Stochastic Optimization under Confounding Effect"
_ICLR.cc/2025/Conference — Submitted to ICLR 2025_

### Official Review · Reviewer_XxxQ · 2024-11-02

**Soundness:** 3
**Presentation:** 2
**Contribution:** 3
**Rating:** 6
**Confidence:** 3

**Summary:**

This paper looks at the problem of policy learning/decision making under confounding, and in particular, when the some (potentially high dimensional) proxy ($Z$) instead of the confounding variables ($X$) are observed. Since the proxy might not capture the full information from the confounders, this can be considered as partial-information regime, and will have some excess regret when compared to the full-information regime. Under exponential assumptions on the data generating process of the proxy, one may infer the underlying $X$ from the $Z$. The authors builds on a weighted sample average approximation, and propose to use the recovered $X$ to learn the optimal decision. The paper provides non-convergent regret bounds under confounding effects and shows that the proposed method improves this bound.

**Strengths:**

I appreciate the explanations about each assumptions on how on whether they are standard/commonly used, the potential implications, and the role it plays in the proofs of the technical results. Moreover, the idea of framing the problem as matrix completion to recover confounding variables from observed proxies. The theoretically results also suggest the advantages of the proposed approach.

**Weaknesses:**

- The current discussion on the background (e.g. the detailed example on uncondoundedness, and discussion on the distinction between) might be too detailed for the main paper. It might be better to move parts of them to the appendix, and leave room for more discussion on the theoretical results of the proposed method.
- Some of the definitions can be presented in a more organized way. For instance, in assumption 4, $\mathbf{B}^*$ is referenced before defining might be better to include any definitions in the assumptions.
- I find section 3 to be a bit unclear. For instance, when learning the M-estimator, is $\hat{r}$ also a learned parameter or is it choosen? How does it relate to the rank of $V$? Since $\hat{r}$ also appears in the regret bound, it would be nice to include a discussion on $\hat{r}$ to understand how much of the excess regret is due to limitations of the estimation approach and how much is due to the intrinsic nature of the data generating process (e.g. low rank $V$).

**Questions:**

Throughout the paper, it was explained that if $Z$ captures all the information from $X$, is there a concrete/mathematical way quantify this from the data generating process? This might be helpful for decomposing the regret into irreducible parts (due to the data generating process) and what arises from suboptimality of the algorithm (if any).

---

> ### Author Response · Authors · 2024-11-25
> **Response to Reviewer XxxQ by Authors (1/2)**
>
> Thanks for your positive evaluation of the article and the valuable suggestions provided. Your feedback is constructive and helpful in improving and enhancing the paper.  Please find below the detailed reply to your comments.
>
> **Q1: The current discussion on the background (e.g. the detailed example on uncondoundedness, and discussion on the distinction between) might be too detailed for the main paper. It might be better to move parts of them to the appendix, and leave room for more discussion on the theoretical results of the proposed method.**
>
> Thank you for your valuable suggestion. It has been very helpful in improving the writing of the paper. We have moved the detailed descriptions of unconfoundedness from the Introduction and model sections to a new Appendix D. For example, the explanation of Assumption 1 in the second paragraph on page 4 has been moved there. Additionally, we have provided more discussion on the theoretical results. For instance, we have expanded the explanation and discussion of $\hat{r}$, as you mentioned in the third question.
>
> **Q2: Some of the definitions can be presented in a more organized way. For instance, in assumption 4, $B^∗$ is referenced before defining might be better to include any definitions in the assumptions.**
>
> We thank the reviewer for the suggestion to improve the paper's readability.  We have reorganized the presentation of some definitions to improve clarity. For example, we have moved the introduction of the decomposition of the decision $p$ to before Assumption 2. Moreover, we have now introduced the definition of $B^{*}$ within Assumption 4.
>
> **Q3: I find section 3 to be a bit unclear. For instance, when learning the M-estimator, $\hat{r}$ is also a learned parameter or is it choosen? How does it relate to the rank of $V$?**
>
> We apologize for not clearly explaining the meaning of $\hat{r}$. $\hat{r}$ represents the estimated rank of the matrix A, which is determined through cross-validation. Additionally,
> $\hat{r}$ is also the dimension of the estimated confounders in our problem, which is equivalent to the number of columns in matrix $X$ and the number of rows in matrix $V$. Therefore, $\hat{r}$ serves as an upper bound for the rank of matrix $V$.
>
> **Since $\hat{r}$ also appears in the regret bound, it would be nice to include a discussion on $\hat{r}$ to understand how much of the excess regret is due to limitations of the estimation approach and how much is due to the intrinsic nature of the data generating process (e.g. low rank).**
>
>  Thank you for your valuable suggestion. Regret is indeed caused by the limitations of the estimation approach and the intrinsic nature of the data generating process. In Theorem 3 and 4, the regret bounds in Equations (15) and (17) are divided into two parts. The first part is due to errors caused by non-parametric estimation, while the second part is attributed to errors resulting from estimating confounders under the low-rank data structure and Assumption 3 proposed data generating process.
>
> We have further explored the significant and multifaceted impact of $\hat{r}$ on regret and incorporated additional analysis in the paper. Our findings show that the choice of $\hat{r}$ is crucial for both sources of regret.
>
> In Theorem 3, where the true dimension of confounders is known, $\hat{r}$ equals $m$. The analysis in Theorem 4 reveals that selecting an appropriate $\hat{r}$ can enhance convergence speed for the regret caused by non-parametric estimation. In cases where the true dimension of confounders is unknown, the regret caused by data generating process will not converge due to the discrepancy between $x$ and $\hat{x}$. Nevertheless, an optimal choice of $\hat{r}$ can mitigate the regret in the second part. Specifically, aligning the value of $\hat{r}$ with the dimension of confounders, as shown in Theorem 4, leads to convergence of the second part to 0, underscoring the critical role of $\hat{r}$ in the algorithm performance.
>
> We have also included an analysis of $\hat{r}$'s impact on performance in both synthetic and real data experiments, demonstrating that the algorithm performs better when $\hat{r}$ approximates the true dimension of confounders, supporting the theoretical findings. We discovered this result during the process of cross-validation, hence it was not detailed in the submitted manuscript.

---

> ### Author Response · Authors · 2024-11-25
> **Response to Reviewer XxxQ by Authors (2/2)**
>
> **Q4: Throughout the paper, it was explained that if $Z$ captures all the information from $X$, is there a concrete/mathematical way quantify this from the data generating process? This might be helpful for decomposing the regret into irreducible parts (due to the data generating process) and what arises from suboptimality of the algorithm (if any).**
>
> The amount of confounder information contained in the proxy features does indeed directly impact the algorithm's performance. In Theorem 2, we decompose the regret into two parts, where the first part is caused by the sub-optimality of the algorithm (which converges to 0), and the second part is due to the discrepancy between proxy features and confounders caused by the data generating process. In Equation (10), if the dimensions of the two are the same, the regret caused by the data generating process can be quantified as the Euclidean distance between proxy features $z$ and confounders $x$.
>
> In the performance of our proposed algorithm, the first part of Equations (15), (16), and (17) still represents the regret caused by the sub-optimality of the algorithm. The second part quantifies the regret caused by the data generating process, which depends on the discrepancy between the estimated confounders and the true confounders in containing the information of uncertainty. In Equations (15) and (17), this quantified regret can converge to 0 using the method we propose.
>
> Therefore, although we did not directly quantify the situation where "$Z$ captures all confounders information" in the paper, we decomposed regret into two parts related to algorithm sub-optimality and data generating process, where the magnitude of the second part can be quantified by the difference between $Z$ and $X$.

---

> ### Comment · Reviewer_XxxQ · 2024-11-28
>
> Thank you for the detailed discussion, I will keep my score.

---

### Official Review · Reviewer_kye8 · 2024-11-02

**Soundness:** 2
**Presentation:** 2
**Contribution:** 2
**Rating:** 3
**Confidence:** 3

**Summary:**

This paper studies the confounding effects and endogeneity in decision-making. Several stochastic regret bounds are derived to quantify the impact of confounding effects. Then a semi-parametric decision-making framework based on proxy matrices is proposed, aiming at addressing stochastic optimization problems under confounding effects. Experiments on real dataset shows the proposed framework can mitigate the repercussions arising from the unmet assumption of unconfoundedness.

**Strengths:**

1、The problem of confounding effects and endogeneity in making  decisions is important.

2、The proofs of the theorems of regret bounds are good.

3、Experiments show some effectiveness of the proposed decision-making framework on real dataset.

**Weaknesses:**

1、Proposed framework relies on too many theoretical assumptions. The regret bound may be highly sensitive to certain assumptions, where even slight deviations could lead to significant changes in outcomes. I thereby questioning the robustness of the theory.

2、In the decision-making framework, why can we assume the linearity between $X$ and $Z$? Under this assumption, when $n$ is high dimensional, how to handle with the overfitting problem?

3、The proposed algorithm is not clear. From the decision-making framework, I assumed the Weighted Sample Average Approximation is the product of $\mathbf{\hat{X}}$ and $\hat{V}$, then what will happen when the given $\mathbf{\hat{X}}$ and $\hat{V}$ differs with the actual \mathbf{X}$ and $V$? More theoretical analysis about this is needed.

4、The experimental results are weak. Only one real dataset is used in experiments, limited data may not adequately test the robustness of the model under different conditions.

**Questions:**

Please response the questions in weaknesses.

---

> ### Author Response · Authors · 2024-11-25
> **Response to Reviewer kye8 by Authors (1/3)**
>
> We thank the reviewer for the constructive comments and suggestions, we elaborated more on your concerns below, please have a look for more information.
>
> **Q1: Proposed framework relies on too many theoretical assumptions. The regret bound may be highly sensitive to certain assumptions, where even slight deviations could lead to significant changes in outcomes. I thereby questioning the robustness of the theory.**
>
> We would like to point out that the theoretical guarantees of the proposed method rely on mild assumptions commonly found in the literature and do not affect the robustness of the results. We have not made any assumptions about the form of uncertainty, only general requirements for the utility function's continuity. The generation process of proxy can produce various types of data and scenarios observed in reality. The technical assumptions in Assumption 4 only require the stability of the matrices, which can be verified in real-world data. Therefore, the model is applicable to different uncertainty distributions, various forms of utility functions, and observed proxy features in reality, and these assumptions do not affect the model's robustness. This is also confirmed in experiments with real data, where our model outperforms other methods on different real datasets and various dimensions of proxy features. Below, we provide a more detailed explanation of each assumption.
>
> Assumption 1 is necessary in the literature for studying causal effects or stochastic optimization with endogeneity, and only Theorems 1 and 2 in our paper require this assumption. Our proposed decision framework aims to address the implications that arise when Assumption 1 cannot be met, so the main theoretical results, Theorems 3 and 4, do not require Assumption 1.
>
> Assumption 2 entails assumptions about the feature space and objective function of the stochastic optimization problem, similar assumptions that are commonly employed in related literature. The feature space must have finite support, and the objective function needs to satisfy some mild continuity requirements. In general, when estimating the objective function using data, the function must possess certain fundamental properties to ensure the accuracy of data-driven approximation.
>
> Assumption 3 pertains to the assumption about the generation process of proxy features. We claim that the generation process of proxy features is not restrictive. First, even if there exists a nonlinear relationship between the proxy features and the confounders, the general relationships can be approximated by a linear form with a much larger feature space by Taylor approximation. Second, we do not impose restrictions on the distribution $\mathbb{P}(\cdot)$, which enhances the robustness of the structure of the proxy features. Different distributions can also generate proxy features of different data types. Third, the generative process in Section 3.1 can also explain some special cases in reality: (i) if the observed $j$th proxy feature is exactly the $s^\prime$th confounder, then this implies $v_{s^\prime j}=1$ and $v_{sj}=0$ for $s\neq s^\prime$; (ii) if the $j$th proxy feature is a one-to-one noisy realization of the $s^\prime$th confounder, then this implies $v_{sj}=0$ for $s\neq s^\prime $; (iii) if a proxy feature is unrelated to any of the confounders, then it is directly generated by $\mathbb{P}(\cdot|0)$.
>
> Assumption 4 pertains to the properties of matrices. Assumption 4(i) setting a low-rank condition is a prerequisite for matrix completion methods. Firms often observe numerous proxy features to represent a smaller number of confounders, implying that the matrix $A$ defined in data generating process is low-rank. From a theoretical standpoint, sufficiently large matrices obtained from certain latent variable models are approximately low-rank.
> Assumption 4(ii)-(iv) indicate that matrices $A$ and $X$ need to exhibit a certain level of stability. If the underlying confounders matrix $X$ or the proxy matrix $A$ before noisy realization are highly unstable, for instance, if there are extreme values present, then the difficulty of estimation will increase, and the precision of the estimates will be harder to control. Therefore, assuming stability for these two matrices is a reasonable and commonly used technical setup to control the theoretical performance of the estimates. We also want to claim that the stability assumption is reasonable in practice. For example, in the pricing problem studied in our experiments, the features of products (i.e., confounders) in the data are unlikely to undergo significant changes, as the features of the same product or similar products are generally quite similar. Additionally, the proxy features from different observation points do not fluctuate greatly, and in many cases, they are even the same, such as certain price-related features, weather-related features, discount-related features, and so on.

---

> ### Author Response · Authors · 2024-11-25
> **Response to Reviewer kye8 by Authors (2/3)**
>
> **Q2: In the decision-making framework, why can we assume the linearity between  $X$ and $Z$?**
>
> The relationship between $Z$ and $X$ is $Z~P(·|A)$, where $A$ can be viewed as a de-noised proxy matrix and we assume that there is a linear relationship between the confounders matrix $X$ and matrix $A$. Therefore, there is no direct relationship between the proxy matrix $Z$ and matrix $X$, and the proxy matrix $Z$ is not obtained through a simple linear transformation. As mentioned in our response to the first question, the assumption of a linear relationship between matrices $A$ and $X$ is generalizable. It can be used to explain both common and special cases when observing proxy features in reality. Furthermore, if there exists a nonlinear functional relationship between $A$ and $X$, we can approximate it using Taylor expansion, for example, if $z_j=f(x_j)$ where $x_j=(x_{1j},x_{2j},....x_{mj})$. After using Taylor expansion, we can obtain $z_j=f(0)+\sum_{s=1}^{m} \partial f_{s}(0) x_s+\sum_{s=1}^{m}\sum_{t=1}^{m} [\partial^2 f(0)]_{st} x_s x_t+...+\epsilon$. This function is still a linear function, with the feature dimension changing from $m$ to $m+m^2$.
>
> In conclusion, due to the approximate linearity, interpretability, and strong theoretical guarantees, we assumed a linear relationship between $X$ and $A$. However, we can also relax this assumption to a non-linear relationship and utilize kernel methods to handle this non-linearity.
>
> **Under this assumption, when $n$ is high dimensional, how to handle with the overfitting problem?**
>
> Under Assumption 3, if the dimension of confounders is too high, overfitting can indeed occur. We employed a nuclear norm regularization in Equation (11) to avoid overfitting. The level of regularization is determined by the tuning parameter $\lambda$. As discussed below Equation (11), a higher $\lambda$ value corresponds to a lower estimated rank of matrix $A$, indicating a lower dimensionality of the confounders in the estimation process. Controlling the dimensionality of the confounders helps mitigate the issue of overfitting.
>
> **Q3: The proposed algorithm is not clear. From the decision-making framework, I assumed the Weighted Sample Average Approximation is the product of $\hat{X}$ and $\hat{V}$, then what will happen when the given \hat{X} and \hat{V} differs with the actual $X$ and $V$? More theoretical analysis about this is needed.**
>
> Our algorithm consists of two main steps. In the first step, we use M-estimation to estimate the confounders, resulting in estimated values of the confounders matrix $\hat{X}$ and loading matrix $\hat{V}$. The estimated matrix $\hat{X}$ is then utilized in the next step, which involves a weighted Sample Average Approximation (wSAA) framework based on the estimated confounders matrix $\hat{X}$, leading to the final decision value of the algorithm.
>
> wSAA is an extension of the Sample Average Approximation (SAA) method used in optimization under uncertainty. In wSAA, each sample in the averaging process is assigned a weight based on its importance or relevance. In our problem, $X$ contains $n+1$ rows, with the first $n$ rows representing estimates of confounders for $n$ historical data points, and the last row representing estimates of current confounders. The weight assigned to each historical data point depends on the correlation between the current scenario and historical scenarios. This correlation is determined by the distance between the combination of estimated confounders and the decision (i.e., ($\hat{x}$,$q$)) and the combination of the $i$th estimated historical confounders and historical decision (i.e., ($\hat{x_i}$,$q_{i}$)). When using the kNN weight function, the data points corresponding to the $k$ closest vectors ${(\hat{x_t},q_t)}_{t=1,...,k}$ to the vector ($\hat{x}$,$q$) are assigned a weight of $1/k$, while the weight for other data points is 0. Based on the calculated weights, we average the historical utility experiences to solve for the optimal decision.
>
> As you pointed out, the deviation between $\hat{X}$ and the true $X$ can impact the theoretical performance of the algorithm. This impact is demonstrated in Theorems 3 and 4. The deviation between $\hat{X}$ and $X$ results in regret in the second part of the right-hand side of Equations (15), (16), and (17).

---

> ### Author Response · Authors · 2024-11-25
> **Response to Reviewer kye8 by Authors (3/3)**
>
> **Q4: The experimental results are weak. Only one real dataset is used in experiments, limited data may not adequately test the robustness of the model under different conditions.**
>
> We agree with your point, so we have supplemented the experiments. Our real dataset is sourced from our collaborative partner, AEON, one of the largest retail companies in Japan. The original dataset comprises 171,709 sales records from four stores, spanning from August 1st to August 31st, 2023. It includes 712 features such as product category, weather-related variables, sales-related metrics, pricing information, discount-related factors, among others. For the current experiment, we selected sales data from one store for the week of August 1st to August 7th for preprocessing and experimentation. Our dataset encompasses data from different stores and time periods, thus representing various contexts. To assess the model's robustness, we conducted experiments using data from another store from August 8th to August 14th. We found that the main conclusions of the experiments remained consistent across different datasets, and detailed results are presented in Appendix C.

---

### Official Review · Reviewer_nLWY · 2024-11-03

**Soundness:** 2
**Presentation:** 2
**Contribution:** 2
**Rating:** 3
**Confidence:** 2

**Summary:**

This paper addresses the challenge of quantifying the impact of confounding effects in real-world decision-making, where high-dimensional, heterogeneous proxy features of confounders are often used. To tackle this, the authors propose a novel semi-parametric decision framework that combines exponential family matrix completion to estimate a confounders matrix from these proxy features. This estimated matrix is then used to make more accurate and reliable decisions by accounting for confounding effects in the data.

**Strengths:**

1. This paper addresses a realistic and relevant problem for real-world applications: quantifying the impact of confounding effects.

2. The paper is well-written and grounded in theory, with proofs for all claims included in the appendix.

**Weaknesses:**

1. The experiment uses proxy features to estimate confounders, but proxy features are often high-dimensional and heterogeneous, which may not fully represent the true confounders. I am interested in how the ability of the proxy variables affects proposed methods. I assume  if the quality and representativeness of the proxy features are insufficient, it could introduce bias into the estimated confounders matrix.


2. The paper currently lacks a discussion on related work. In the field of causal effect estimation, there is extensive research on recovering confounders from proxy variables. I highly recommend including a literature review that addresses these studies. To further validate the proposed method, I suggest designing an experiment that compares the performance of two approaches: (1) representation learning directly from proxy variables, and (2) using estimated confounders matrices obtained from proxy variables. [1]

[1] Causal Effect Inference with Deep Latent-Variable Models in Nips 2017.

**Questions:**

See Above.

---

> ### Author Response · Authors · 2024-11-25
> **Response to Reviewer nLWY by Authors (1/2)**
>
> We thank the reviewer for the constructive comments. We have revised the manuscript thoroughly to take your comments into account. All the modifications to the manuscript are highlighted in blue. Please find below the detailed reply to your comments.
>
> **Q1: The experiment uses proxy features to estimate confounders, but proxy features are often high-dimensional and heterogeneous, which may not fully represent the true confounders. I am interested in how the ability of the proxy variables affects proposed methods. I assume if the quality and representativeness of the proxy features are insufficient, it could introduce bias into the estimated confounders matrix.**
>
> Thank you for your insightful comments. Your point is valid; if the quality and representativeness of the proxy features are insufficient, it could introduce bias into the estimated confounders matrix and further affect the performance of the decisions. We can analyze this issue from both theoretical and experimental perspectives.
>
> From Theorems 3 and 4, it can be observed that the second part of the regret bound (or the performance of the proposed method) depends on the discrepancy between the estimated $\hat{x}$ and the true $x$. Based on the specific form of the bound for the second part, we find that the bias of the estimated confounders $\hat{x}$ depends on the complex relationship among the number of historical observations $n$, the dimension of proxy features $b$, and the total number of observed features $|\Omega|$. Theoretically, these parameters determine the quality and representativeness of proxy features. Based on theoretical results, we can interpret $\frac{|\Omega|}{r^3 m^2 \log m}$ in Theorem 3 and $\frac{|\Omega|}{r^5 b^2 \log b}$ in Theorem 4 as indicators of the quality or representativeness of proxy features. Obviously, higher quality leads to lower regret bound and better performance of the proposed model.
>
> Our experimental results also validate the above points. In synthetic data experiments, a higher dimension of proxy features contains more information about confounders. Therefore, in Figure 3, we observe that with a fixed number of data points $n$, as the dimension of proxy features (i.e., the value of $b$) increases, the out-of-sample profit of the PMFI-model also increases. This finding holds true in real data experiments as well. In Figure 7 of Appendix C, a larger value of $b$ implies higher quality of proxy features as it includes more information about confounders. We can also find that better quality of proxy features leads to improved performance of our proposed method.

---

> ### Author Response · Authors · 2024-11-25
> **Response to Reviewer nLWY by Authors (2/2)**
>
> **Q2: The paper currently lacks a discussion on related work. In the field of causal effect estimation, there is extensive research on recovering confounders from proxy variables. I highly recommend including a literature review that addresses these studies.**
>
> Thank you for your valuable suggestions, which have greatly enhanced the completeness of our paper and clarified its position in the literature. In the field of causal effect research, various methods have been proposed to handle confounding effects.
>
> In the field of causal effect research, various methods have been proposed to handle confounding effects. For instance, instrumental variables (IV) are widely used to estimate causal effects (Carrasco et al., 2007; Baiocchi et al., 2014; Chen & Qiu, 2016; Guo & Small, 2016; Mogstad & Torgovitsky, 2018). Chen et al. (2016) explore the use of IVs in nonparametric and semiparametric regression models to address endogeneity issues. Another approach focuses on estimating causal effects using proxy features. The idea is to first understand the distributional relationship between confounders and proxies, adjust the confounders, and then identify causal effectss (Wooldridge, 2009; Pearl, 2012; Cai & Kuroki, 2012; Kuroki & Pearl, 2014; Edwards et al., 2015; Miao et al., 2018; Tchetgen et al., 2020). Studies. For example, Tchetgen et al. (2020) propose the proximal g-computation algorithm to adjust the influence of unmeasured confounding through the use of proxy variables. Studies most closely related to our approach involves inferring confounders using observed proxies, typically based on latent-variable models ls (Kingma, 2013; Louizos et al., 2017; Kallus et al., 2018). Variational Autoencoders (VAE) (Kingma and Welling, 2013) is a highly effective heuristic method, based on which Louizos et al. (2017) propose the Causal Effect VAE, which can simultaneously estimate unknown latent spaces summarizing confounding factors and estimate causal effects. In more complex data scenarios, other methods for inferring hidden confounders have also been developed (Guo et al., 2020; Chu et al., 2021; Ma et al., 2021).
>
> Due to the limitations of space in the article, we have included a concise version of the above overview in the revised manuscript. We believe that some techniques for understanding confounders nested within causal effects also have the potential to be combined with stochastic optimization problems. These methods for studying causal effects can also serve as intermediate steps for our stochastic optimization. However, these techniques require additional assumptions about the problem and lead to completely different decision frameworks. This is beyond the scope of our study.
>
> The most relevant literature to our research includes studies proposing methods for estimating confounders, such as VAEs, Balancing Neutral Networks (BNN), and others. However, it should be noted that these methods are often black-box or heuristic and cannot provide theoretical guarantees or insights into the confounding effect. The decision framework proposed in this paper addresses confounding effects while providing ample theoretical support and interpretability to assist decision-makers in making better decisions. Additionally, in supplementary experiments, our proposed method has been shown to outperform classic or advanced methods for confounder estimation.
>
> **To further validate the proposed method, I suggest designing an experiment that compares the performance of two approaches: (1) representation learning directly from proxy variables, and (2) using estimated confounders matrices obtained from proxy variables.[1]**
>
> Thanks for your very constructive suggestions on improving the effectiveness of the method we proposed. We fully agree with your point. In the numerical experiments, we have added experiments using some representation learning methods. Specifically, we compared our method with the classical Principal Components Analysis (PCA) method, as well as the state-of-the-art methods VAEs and BNN. The experimental results have been updated in Table 1. We found that our method outperforms the above-mentioned methods on real datasets. Furthermore, we want to emphasize that our method can provide theoretical guarantees to guide decision-makers and offer insights.

---

### Official Review · Reviewer_hzBV · 2024-11-04

**Soundness:** 3
**Presentation:** 3
**Contribution:** 2
**Rating:** 5
**Confidence:** 3

**Summary:**

The paper introduces a semi-parametric decision-making framework that combines exponential family matrix completion with weighted sample average approximation (wSAA) to estimate the confounders matrix from high-dimensional proxy features and use it for decision-making. The authors provide theoretical analysis and experimental results of the framework.

**Strengths:**

1. The paper is generally easy to follow. The technical results seem sound to me.
2. The authors clearly state the limitations of previous results based on the non-confounding assumption and motivate the use of proxy matrix.
3. The authors also conduct experiments on real dataset.

**Weaknesses:**

While this paper proposes a framework for mitigating confounding effects in contextual optimization, I am uncertain about the overall significance of its contributions. Several aspects warrant further exploration:

- **Comparison with Existing Proxy Methods:** A substantial body of literature addresses confounding issues using proxies, notably through methods such as instrumental variables (IV) (Chen et al, 2016, Carrasco et al, 2007) and negative control proxies (Tchetgen et al, 2020). I believe a thorough comparison with the above-mentioned methods are necessary.

- **Choice of kNN and High-Dimensionality Concerns:** The framework’s reliance on k-nearest neighbors (kNN) raises concerns regarding its suitability for high-dimensional settings, as kNN gives a slow statistical rate $n^{-1/(2p + 2\hat r)}$. This is particularly relevant in high-dimensional contexts, where faster rates are often achievable through minimax estimators (Dikkala et al, 2020) and also in the context of contextual policy optimization (Chen et al, 2023).

- **Technical Novelty:** The proposed framework seems to primarily combine matrix completion or factorization with a wSAA framework, both well-studied methods. The paper does not clearly highlight any novel technical advancements beyond this combination, making it difficult to identify the methodological contributions.

- **Assumptions and Practical Limitations:** Also, I found some assumptions to be quite strong. For instance, Assumption 3 on the strong convexity of the log partition function, and Assumption 4 (iv) on the estimation accuracy of $\hat x$. To my understanding, there are cases where the underlying confounder $x$ is not identifiable. These assumptions may limit the general applicability of the framework and the authors do not make enough effort to justify these assumptions.


***References***
1. Chen, X., & Qiu, Y. J. J. (2016). Methods for nonparametric and semiparametric regressions with endogeneity: A gentle guide. Annual Review of Economics, 8(1), 259-290.
2, Carrasco, M., Florens, J. P., & Renault, E. (2007). Linear inverse problems in structural econometrics estimation based on spectral decomposition and regularization. Handbook of econometrics, 6, 5633-5751.
3. Tchetgen, E. J. T., Ying, A., Cui, Y., Shi, X., & Miao, W. (2020). An introduction to proximal causal learning. arXiv preprint arXiv:2009.10982.
4. Dikkala, N., Lewis, G., Mackey, L., & Syrgkanis, V. (2020). Minimax estimation of conditional moment models. Advances in Neural Information Processing Systems, 33, 12248-12262.
5. Chen, S., Wang, Y., Wang, Z., & Yang, Z. (2023). A unified framework of policy learning for contextual bandit with confounding bias and missing observations. arXiv preprint arXiv:2303.11187.

**Questions:**

I didn't find the reliable source of the real-world dataset, while the authors should provide more information about the dataset used for evaluating the proposed method.

**Details Of Ethics Concerns:**

No concerns

---

> ### Author Response · Authors · 2024-11-25
> **Response to Reviewer hzBV by Authors (1/3)**
>
> Thanks for your positive evaluation and valuable suggestions for the paper.  Please find below the detailed reply to your comments.
>
> **Q1: Comparison with Existing Proxy Methods: A substantial body of literature addresses confounding issues using proxies, notably through methods such as instrumental variables (IV) (Chen et al, 2016, Carrasco et al, 2007) and negative control proxies (Tchetgen et al, 2020). I believe a thorough comparison with the above-mentioned methods are necessary.**
>
> Thank you for your suggestions, which have been very helpful in enhancing the completeness of our paper and clarifying its positioning in the literature. We reviewed literature on addressing confounding effects in causal effect studies, and specifically compared our method with instrumental variable methods and methods using proxy features to infer confounders, thus defining the position of our study more clearly in the literature. Additionally, within the scope of our study, we compared our proposed method with classical and state-of-the-art representation learning methods on real datasets, further enhancing the comparison between our proposed method and existing methods in the literature.
>
> In the field of causal effect research, various methods have been proposed to handle confounding effects. For instance, instrumental variables (IV) are widely used to estimate causal effects (Carrasco et al., 2007; Baiocchi et al., 2014; Chen & Qiu, 2016; Guo & Small, 2016; Mogstad & Torgovitsky, 2018). Chen et al. (2016) explore the use of IVs in nonparametric and semiparametric regression models to address endogeneity issues.
> Another approach focuses on estimating causal effects using proxy features. The idea is to first understand the distributional relationship between confounders and proxies, adjust the confounders, and then identify causal effectss (Wooldridge, 2009; Pearl, 2012; Cai & Kuroki, 2012; Kuroki & Pearl, 2014; Edwards et al., 2015; Miao et al., 2018; Tchetgen et al., 2020). Studies. For example, Tchetgen et al. (2020) propose the proximal g-computation algorithm to adjust the influence of unmeasured confounding through the use of proxy variables.
> Studies most closely related to our approach involves inferring confounders using observed proxies, typically based on latent-variable models ls (Kingma, 2013; Louizos et al., 2017; Kallus et al., 2018). Variational Autoencoders (VAE) (Kingma and Welling, 2013) is a highly effective heuristic method, based on which Louizos et al. (2017) propose the Causal Effect VAE, which can simultaneously estimate unknown latent spaces summarizing confounding factors and estimate causal effects. In more complex data scenarios, other methods for inferring hidden confounders have also been developed (Guo et al., 2020; Chu et al., 2021; Ma et al., 2021).
>
> It should be noted that the techniques proposed by Chen et al. (2016) and Tchetgen et al. (2020) for addressing endogeneity or confounding effects using instrumental variables and proxy features are closely related to parameter estimation or causal effect estimation. We believe that these techniques can also improve the performance of stochastic optimization problems with existing confounding effects. However, this is beyond the scope of our study. As discussed above, the most relevant approach or literature to the current paper involves inferring confounders using proxy features. In order to provide a more comprehensive comparison with existing methods and literature, we have conducted additional experiments. These experiments utilize some advanced or classic methods for estimating confounders from proxy features, such as Variational Autoencoders, Principal Component Analysis, Balancing Neural Network, and other representation learning methods, and compare them with the proposed method in the paper.

---

> ### Author Response · Authors · 2024-11-25
> **Response to Reviewer hzBV by Authors (2/3)**
>
> **Q2: Choice of kNN and High-Dimensionality Concerns: The framework’s reliance on k-nearest neighbors (kNN) raises concerns regarding its suitability for high-dimensional settings, as kNN gives a slow statistical rate $n^{-\frac{1}{2p+2\hat{r}}}$. This is particularly relevant in high-dimensional contexts, where faster rates are often achievable through minimax estimators (Dikkala et al, 2020) and also in the context of contextual policy optimization (Chen et al, 2023).**
>
> Thanks for your insightful evaluation. The convergence rate of our proposed method when using the kNN weight function is closely related to the dimensionality of the features. When the feature dimension is large, the convergence speed does indeed slow down. This is determined by the convergence rate of non-parametric local machine learning. We have not made any assumptions about uncertainty or the form of decision-making, but have only made mild assumptions about the continuity of the utility function. This aligns with real-world scenarios, such as pricing problems, where the form of product demand may be entirely unknown. Due to the mild assumptions on uncertainty distribution information, the proposed model's convergence rate is influenced by the feature dimension. Nevertheless, under the current problem setting, our proposed method has already shown improvement in convergence speed compared with the traditional non-parametric decision method without any assumptions on uncertainty and decision structure, as the value of $\hat{r}$ is relatively small and controllable.
>
> The minimax estimators proposed by Dikkala et al. (2020) and the policy optimization algorithm proposed by Chen et al. (2023) can achieve convergence rates independent of feature dimension under certain assumptions. However, it is important to note that the superior convergence rates depend on specific assumptions. For example, Dikkala et al. (2020) require assumptions that the estimates lie in Reproducing Kernel Hilbert Spaces or high-dimensional linear function classes. On the other hand, the method proposed by Chen et al. (2023) necessitates assumptions about the structure of uncertainty and that the estimates lie in specific function spaces. Therefore, to achieve better convergence rates, additional assumptions are needed, which are beyond the scope of this paper.
>
> It is important to note that the decision framework we propose exhibits strong robustness and is not limited to using the wSAA within that framework. If we set the decision function in Reproducing Kernel Hilbert Spaces or linear function classes, instead of using the wSAA framework, we can anticipate that under some additional assumptions on the uncertainty structure, our proposed decision framework will achieve a regret-bound convergence rate independent of the feature dimension. However, the original intention of this study is not to make any form of assumptions about the distribution of uncertainty.
>
> **Q3: Technical Novelty: The proposed framework seems to primarily combine matrix completion or factorization with a wSAA framework, both well-studied methods. The paper does not clearly highlight any novel technical advancements beyond this combination, making it difficult to identify the methodological contributions.**
>
> We agree that we should have discussed our technical contribution more clearly, we want to emphasize that our decision framework and theoretical results are not simply a combination of matrix completion and wSAA. Our technical contributions are threefold.
>
> 1. Under the premise of complete unknown uncertainty form, we provide an exact regret bound for stochastic optimization problems with endogeneity. This regret bound can be extended to decision frameworks using other local machine learning methods (such as random forest).
>
> 2. We theoretically decompose the regret bound into a part related to non-parametric decision-making and a part determined by the confounding effect, quantifying the confounding effect. This decomposition facilitates and motivates the use of matrix-related methods to enhance the regret caused by the confounding effect.
>
> 3. The technical decomposition of the regret bound allows us to focus solely on the regret bound caused by the confounding effect. The advantages of matrix completion methods, such as their ability to handle heterogeneous data and missing values, as well as their weaker theoretical assumptions, make them a premium choice for addressing the confounding effect. Technically, we combine the impact of the confounding effect with the average error of matrix completion and derive the regret bound of the proposed method in different real-world scenarios.
>
> Overall, the semi-parametric decision framework we propose is an organic combination of decision methods and matrix methods under mild model assumptions. This method simultaneously possesses practical applicability, strong theoretical guarantee, interpretability, and technological innovation.

---

> ### Author Response · Authors · 2024-11-25
> **Response to Reviewer hzBV by Authors (3/3)**
>
> **Q4: Assumptions and Practical Limitations: Also, I found some assumptions to be quite strong. For instance, Assumption 3 on the strong convexity of the log partition function, and Assumption 4 (iv) on the estimation accuracy of $\hat{x}$. To my understanding, there are cases where the underlying confounder $x$ is not identifiable. These assumptions may limit the general applicability of the framework and the authors do not make enough effort to justify these assumptions.**
>
> Thanks for your valuable comments. Due to space constraints in the original manuscript, we did not provide detailed explanations for some assumptions. Below, we have further elaborated on the assumptions you mentioned to illustrate that they are not strong.
>
> In Assumption 3, proxy features are observed indirectly via a noisy channel: specifically, via a sample drawn from the corresponding member of the natural exponential family. We claim that the generation process of proxy features is not restrictive. We do not impose specific restrictions on the distribution $\mathbb{P}(\cdot)$, which enhances the robustness of the structure of the proxy features. Many commonly used distributions in practice satisfy the assumption for the strong convexity of the function $G(\cdot)$, such as Gaussian, Bernoulli, Binomial, Poisson, and Exponential. These distributions are sufficient to generate proxy features for various types of data found in reality, such as numerical continuous data, interval continuous data, nominal categorical data, and ordinal nominal data. Therefore, the assumption of strictly convexity for the $G(\cdot)$ function is not restrictive.
>
> Assumption 4(ii)-(iv) indicate that matrices $A$ and $X$ need to exhibit a certain level of stability. If the underlying confounders matrix $X$ or the proxy matrix $A$ before noisy realization are highly unstable, for instance, if there are extreme values present, then the difficulty of estimation will increase, and the precision of the estimates will be harder to control.
>
> Assumption 4(iv) does not make assumptions about the accuracy of the estimate $\hat{x}$ directly; it simply requires that the estimation error of the last row of the matrix $\hat{X}$ does not explode relative to the estimation error of the matrix $\hat{X}$. The errors of $\hat{x}$ and $\hat{X}$ are still allowed to be very large values, and we do not impose any restrictions on the estimation accuracy itself.
>
> We also want to claim that these stability assumptions are reasonable in practice. For example, in the pricing problem studied in the experiments, the features of products (i.e., confounders) in the data are unlikely to undergo significant changes, as the features of the same product or similar products are generally quite similar. Additionally, the proxy features from different observation points do not fluctuate greatly, and in many cases, they are even the same, such as certain price-related features, weather-related features, discount-related features.
>
> Furthermore, it is important to note that Assumption 4(iv) is necessary for obtaining the results in the second part of Theorem 4 (i.e., Equation (17)); Equations (15) and (16) do not require Assumption 4(iv).
>
> We would also like to point out that the above assumptions are technical and not necessarily required to be met in practical applications. In the synthetic and real data experiments, we did not enforce these assumptions on the data, yet the results demonstrate that our proposed model still outperforms all candidate methods.

---

> ### Author Response · Authors · 2024-11-28
> **Response to Reviewer hzBV by Authors (Questions)**
>
> **I didn't find the reliable source of the real-world dataset, while the authors should provide more information about the dataset used for evaluating the proposed method.**
>
> We apologize for not providing detailed information on the real dataset in the submitted manuscript. We have now included the following supplemental information: Our real dataset is sourced from our collaborative partner, AEON, which is one of the largest retail companies in Japan. Our original dataset consists of a total of 171,709 sales records from four stores, covering the period from August 1st to August 31st, 2023. The original dataset includes 712 features, such as product category, weather-related, sales-related, price-related, discount-related features, among others. For the current experiment, we have selected the sales data of products with prices ranging from 0 to 5 from one store for the week of August 1st to August 7th for preprocessing and experimentation.

---

### Meta-Review · Area_Chair_h2Jf · 2024-12-10

**Metareview:**

There is much to appreciate in the contribution, but I still felt unease about the scholarship and framing of the contribution in the literature as a whole. For instance, the literature on proxies is large, and the discussion below seems to acknowledge only some of the more fundamental uses of it without acknowledging work on, for instance, proxy modeling in POMDPs. One early paper in this topic, "Off-Policy Evaluation in Partially Observable Environments" (Tennenholtz et al., AAAI 2020) has several dozen citations by now.

Semi-parametric, doubly-robust, proxy model estimation has been studied from many angles (see "Semiparametric Proximal Causal Inference", Cui et al., JASA 2024, for a recent paper), so I don't think a reliance on kNNs is attractive, particularly it is a odds with the stated goal of solving high-dimensional problems and at a venue such as ICLR, which is particularly motivated by representation learning.

This is a solid technical contribution that has much to be admired and I can see it eventually being published in a near future, but the paper will benefit from more contextualization with the large and diverse literature on proximal causal learning.

**Additional Comments On Reviewer Discussion:**

I acknowledge there was little discussion coming from reviewers, and I read everything in the rebuttal in detail. I believe that my decision was well-informed from the combination of all of these contributions and my own reading of the paper.

---

### Decision · Program_Chairs · 2025-01-22

Reject